# PepCompass: Navigating Peptide Embedding Spaces Using Riemannian Geometry

**Marcin Możejko**[* 1]  **Adam Bielecki**[* 1]  **Jurand Prądzyński**[1]  **Hyun-Su Lee**[2 3 4 5]  **Antoni Janowski**[1]
**Michal Kmicikiewicz**[6 7 8]  **Paulina Szymczak**[6]  **Karol Jurasz**[1]  **Marcin Traskowski**[1]  **Michał Kucharczyk**[1]
**Marcelo Der Torossian Torres**[2 3 4 5]  **Cesar de la Fuente-Nunez**[2 3 4 5]  **Ewa Szczurek**[6 1]

## Abstract

Antimicrobial peptide discovery is challenged by the astronomical size of peptide space and the relative scarcity of active peptides. While generative models provide latent maps of this space, they typically ignore decoder-induced geometry and rely on flat Euclidean metrics, making exploration distorted and inefficient. Existing manifold-based approaches assume fixed intrinsic dimensionality, which fails for real peptide data. We introduce **PepCompass**, a geometry-aware framework based on a **Union of $\kappa$-Stable Riemannian Manifolds** that captures local decoder geometry while maintaining computational stability. PepCompass performs global interpolation via **Potential-minimizing Geodesic Search (PoGS)** to bias discovery toward promising seeds, and enables local exploration through **Second-Order Riemannian Brownian Efficient Sampling** and **Mutation Enumeration in Tangent Space**, which together form **Local Enumeration Bayesian Optimization (LE-BO)**. PepCompass achieves a 100% *in-vitro* validation rate: PoGS

identifies four novel seeds and LE-BO optimizes them into 25 highly active, broad-spectrum peptides, demonstrating that geometry-informed exploration is a powerful paradigm for antimicrobial peptide design.

## 1. Introduction

Efficient exploration of peptide space is notoriously difficult. At the global level, there are more than $3.3 \times 10^{32}$ combinatorially possible amino acid sequences of length at most 25. At the local level, each peptide of length 25 has nearly 1000 neighbors within an edit radius of one. Moreover, only a small fraction of amino acid sequences correspond to antimicrobial peptides (AMPs), which have high activity against bacteria (Szymczak & Szczurek, 2023; Szymczak et al., 2025). This extreme combinatorial complexity renders AMP discovery by brute-force exploration intractable. To address this challenge, we turn to one of the most prolific inventions of humankind: maps.

Since the dawn of civilization, maps have provided a structured way to support both local and global navigation, driving scientific discovery. In modern machine learning, latent-space generative models such as VAEs, GANs, WAEs, and normalizing flows (Bond-Taylor et al., 2022) enable building continuous latent representations—maps—of peptides. Such maps have already facilitated the discovery of promising new AMPs (Szymczak et al., 2023; Oort et al., 2021; Wang et al., 2023; Das et al., 2021). The standard workflow assumes that once the model is trained, its latent space together with the decoder properly models a set of valid, synthetizable peptides. The latent space is typically chosen to be $\mathbb{R}^d$ with a flat Euclidean metric, enabling direct application of existing exploration and optimization algorithms. However, such flat representations suffer from a significant flaw: they ignore the differential geometry induced by the decoder, leading to distortions in distances.

Typical approaches attempting to circumvent this problem assume the *manifold hypothesis* (Bengio et al., 2013) and use the pullback metric (Arvanitidis et al., 2021a). How-

---

[*]Equal contribution  [1]Faculty of Mathematics, Informatics, and Mechanics, University of Warsaw, Warsaw, Poland [2]Machine Biology Group, Departments of Psychiatry and Microbiology, Institute for Biomedical Informatics, Institute for Translational Medicine and Therapeutics, Perelman School of Medicine, University of Pennsylvania, Philadelphia, Pennsylvania, USA [3]Departments of Bioengineering and Chemical and Biomolecular Engineering, School of Engineering and Applied Science, University of Pennsylvania, Philadelphia, Pennsylvania, USA [4]Department of Chemistry, School of Arts and Sciences, University of Pennsylvania, Philadelphia, Pennsylvania, USA [5]Penn Institute for Computational Science, University of Pennsylvania, Philadelphia, Pennsylvania, USA [6]Institute of AI for Health, Hemholtz Center Munich, Neuherberg, Germany [7]School of Computation, Information and Technology, Technical University of Munich [8]Munich Center for Machine Learning. Correspondence to: Ewa Szczurek <ewa.szczurek@helmholtz-munich.de>.

*Proceedings of the $43^{rd}$ International Conference on Machine Learning*, Seoul, South Korea. PMLR 306, 2026. Copyright 2026 by the author(s).

ever, recent work (Brown et al., 2023; Loaiza-Ganem et al., 2024; Wang & Wang, 2024) has shown that the manifold hypothesis does not withstand empirical scrutiny for image data, where sets of images are better modeled as unions or CW-complexes of manifolds with varying low dimensionality (we refer to Appendix A for Related Work). We show that peptide spaces suffer from a similar issue and introduce decoder-derived **Union of $\kappa$-Stable Riemannian Manifolds** $\mathbb{M}^\kappa$, which captures both the complex structure of peptide space and its local geometry, with computational stability controlled by a parameter $\kappa$. Intuitively, we cut the globally distorted map into a set of charts that enable efficient and distortion-free exploration and optimization.

Building upon the union-of-manifolds structure, we introduce **PepCompass**, a geometry-informed framework for peptide exploration and optimization at both global and local levels. At the *global level*, we propose Potential-minimizing Geodesic Search (**PoGS**), which models geodesic curves between known prototype peptides to identify promising seeds for further optimization (Figure 1A). We represent geodesics as energy-minimizing curves in peptide space and augment them with a potential function encoding antimicrobial activity. This biases exploration toward regions not only similar to the starting prototypes but also exhibiting higher activity. In doing so, our method extends standard local analogue search around a single prototype into a bi-prototype, controllable regime.

For *local* search on a single manifold from the family $\mathbb{M}^\kappa$, we designed two geometry-informed approaches: Second-Order Riemannian Brownian Efficient Sampling (**SORBES**) and Mutation Enumeration in Tangent Space (**MUTANG**). SORBES is a provably convergent, second-order approximation of the Riemannian Brownian motion (Schwarz et al., 2023; Herrmann et al., 2023), serving as a Riemannian analogue of local Gaussian search. MUTANG addresses the discrete nature of peptide space by reinterpreting the local tangent space not as continuous vectors but as discrete mutations, directly corresponding to amino-acid substitutions. This reinterpretation provides both interpretability and efficiency, enabling enumeration of a given peptide's neighbours. We further combine SORBES and MUTANG into an iterative **Local Enumeration** procedure that densely populates the neighbourhood of a given peptide with valid, diverse neighbours. Finally, integrating this enumeration with a Bayesian optimization scheme yields an efficient Local Enumeration Bayesian Optimization procedure (**LE-BO**; see Figure 1B,C).

*In vitro* microbiological assays demonstrated unprecedented, 100% success rate of PepCompass in AMP optimization. Using PoGS we derived four peptide seeds, all of which showed significant antimicrobial activity. Further optimization of these seeds with LE-BO yielded 25/25 highly active

peptides with broad-spectrum activity, including activity against multi-resistant bacterial strains. Code is available at `https://github.com/szczurek-lab/pep-compass`.

**Conflict of Interest Disclosure** Merck Healthcare KGaA provides funding for the research group.

## 2. Methods

### 2.1. Background

Let $\mathrm{Dec}_\theta \in C^\infty : \mathcal{Z} \to \mathcal{X}$ be the deterministic decoder mapping latent vectors $z \in \mathbb{R}^d$ to position-factorized peptide probabilities $\mathrm{Dec}_\theta(z) \in \mathbb{R}^{L \times A}$ (with $L$ the maximum peptide length and $A$ the size of the amino acid alphabet $\mathcal{A}$ extended with a padding token $pad$, and $\mathbb{R}^{L \times A}$ - set of matrices of shape $(L, A)$). Define

$$X = \mathrm{Dec}_\theta(\mathcal{Z}) \subset \mathbb{R}^{L \times A},$$

$$\mathrm{p}(z) = \mathrm{argmax}(\mathrm{Dec}_\theta(z), \dim = 1) \in \mathcal{A}^L,$$

where $\mathrm{p}(z)$ denotes the decoded peptide sequence. Intuitively, $X$ is a continuous probabilistic approximation of the peptide space (with $\mathcal{Z}$ as its map), from which the concrete peptides are decoded back using the p operator. For clarity, we drop explicit parameter dependence and simply write $\mathrm{Dec}$ instead of $\mathrm{Dec}_\theta$. We additionally introduce $\hat{\mathrm{Dec}} : \mathcal{Z} \to \mathbb{R}^{LA}$ and $\hat{X} = \hat{\mathrm{Dec}}(\mathcal{Z})$, i.e. the flattened versions of $\mathrm{Dec}$ and $X$.

A common approach is to equip $\mathcal{Z}$ with the standard Euclidean inner product $\langle \cdot, \cdot \rangle_{\mathbb{R}^d}$ and use the associated Euclidean distance as a basis for exploration. This approach is implicitly adopted by many off-the-shelf methods for latent exploration, including usage of K-Means in CMAES (Hansen 2023), Euclidean kernel-based methods in Bayesian optimization (Eriksson & Jankowiak 2021; Hvarfner et al. 2024; Kirschner et al. 2019), as well as standard Gaussian sampling and PCA in Das et al. 2018; Szymczak et al. 2023. However, such approaches ignore the geometry induced by the decoder, which, under the manifold hypothesis, can instead be captured through the *pullback metric* (Bengio et al., 2013; Arvanitidis et al., 2021a).

**Pullback metric** Under the manifold hypothesis, $\hat{\mathrm{Dec}}$ is full rank (i.e. $\mathrm{rank}\, J_{\hat{\mathrm{Dec}}}(z) = d$ for all $z \in \mathcal{Z}$, where $J_{\hat{\mathrm{Dec}}}(z) \in \mathbb{R}^{(LA) \times d}$ is the decoder Jacobian), and $(\mathcal{Z}, G_{\hat{\mathrm{Dec}}})$ is a $d$-dimensional Riemannian manifold (Flaherty & do Carmo, 2013) where $G_{\hat{\mathrm{Dec}}}$ is a natural, decoder-induced pullback metric on $\mathcal{Z}$ given by:

$$G_{\hat{\mathrm{Dec}}}(z) = J_{\hat{\mathrm{Dec}}}(z)^\top J_{\hat{\mathrm{Dec}}}(z) \in \mathbb{R}^{d \times d}, \qquad (1)$$

for $z \in \mathcal{Z}$. To simplify notation, let us set $G_{\mathrm{Dec}} = G_{\hat{\mathrm{Dec}}}$. Now, for a tangent space $T_z\mathcal{Z}$ at a point $z$ and tangent

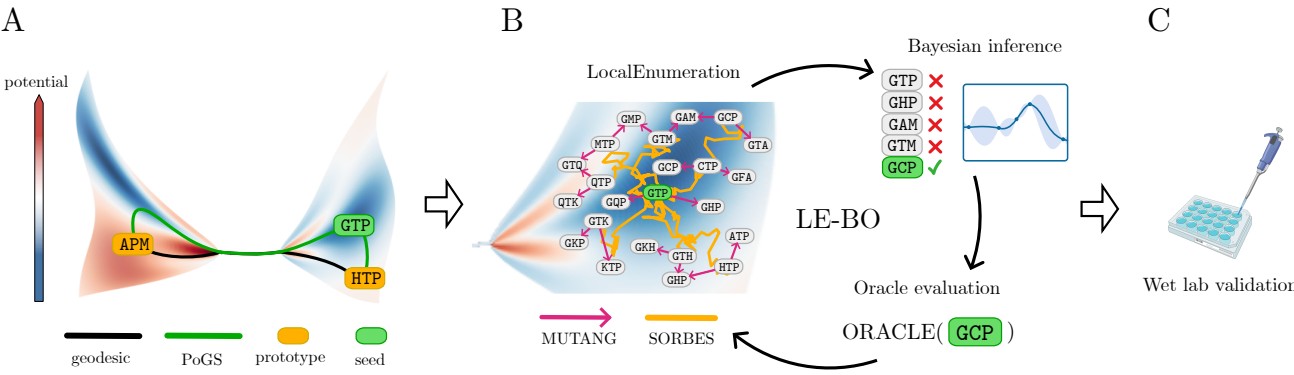

*Figure 1.* **PepCompass overview.** (A) Potential-minimizing Geodesic Search (PoGS) performs global exploration in the decoder-induced latent geometry by constructing potential-augmented geodesic trajectories between prototype antimicrobial peptides. The resulting low-potential regions identify promising seed peptides for downstream optimization. (B) Local Enumeration combines Second-Order Riemannian Brownian Efficient Sampling (SORBES) with Mutation Enumeration in Tangent Space (MUTANG) to generate diverse, geometry-aware local neighborhoods around candidate peptides. Tangent directions induced by the decoder are interpreted as biologically meaningful mutation directions, enabling efficient exploration of valid peptide variants. (C) Local Enumeration Bayesian Optimization (LE-BO) integrates geometry-aware local exploration with surrogate-based Bayesian optimization and oracle evaluation to iteratively discover highly active peptide candidates, which are subsequently validated experimentally.

vectors $u, v \in T_z \mathcal{Z}$,

$$\langle u, v \rangle_z^{\text{Dec}} = u^\top G_{\text{Dec}}(z)\, v. \tag{2}$$

Note that $\hat{\text{Dec}} : (\mathcal{Z}, G_{\text{Dec}}) \to (\hat{X}, \langle \cdot, \cdot \rangle_{\mathbb{R}^{LA}})$ is an isometric diffeomorphism, meaning that distances measured in the latent space locally match distances in ambient space.

**When the manifold hypothesis fails: union of manifolds**
Previous work demonstrates that the manifold hypothesis often fails for complex data such as images (Brown et al., 2023; Loaiza-Ganem et al., 2024; Wang & Wang, 2024), and the data is better represented as unions of local manifolds of varying dimension, typically lower than that of the latent space. However, the previous methods defined the submanifolds based on pre-specified datasets and could not generalize to new data.

### 2.2. Union of $\kappa$-Stable Riemannian Manifolds

Assuming that the manifold hypothesis is indeed violated and that a given generative model has learned to faithfully capture the lower-dimensional structure in the data, it should be reflected in the decoder having rank strictly smaller than the latent dimensionality. We verified this phenomenon for antimicrobial peptides data in two state-of-the-art latent generative models (Das et al., 2018; Szymczak et al., 2023) (see Appendix B). This implies that the $G_{Dec}$ is not of full rank, and consequently the pair $(\mathcal{Z}, G_{\text{Dec}})$ does not constitute a Riemannian manifold.

To address this, we equip each point $z \in \mathcal{Z}$ with a local, potentially lower-dimensional manifold, which we further enrich with a Riemannian structure from the pullback metric. In contrast to previous methods, we adapt a decoder-

dependent approach in the submanifold definition, enabling generalization to any point encoded in the latent space. Namely, we decompose $\mathcal{Z}$ as a union of locally $\kappa$-stable Riemannian submanifolds ($\kappa \geq 0$)

$$\mathbb{M}^\kappa = \{M_z^\kappa : z \in \mathcal{Z}\}, \qquad M_z^\kappa = (W_z^\kappa, G_{\text{Dec}}),$$

where each $W_z^\kappa \ni z$ is an open affine submanifold of $\mathcal{Z}$ of *maximal dimension* (denoted as $k_z^\kappa$ and refered to as $\kappa$−stable dimension), such that the pullback metric $G_{\text{Dec}}$ restricted to $W_z^\kappa$, denoted $G_{\text{Dec}}|_{W_z^\kappa}$, has full rank and satisfies the $\kappa$-stability condition

$$\inf_{\substack{v \in T_z W_z^\kappa \\ \langle v, v \rangle_{\mathbb{R}^d} = 1}} \langle v, v \rangle_z^{\text{Dec}} > \kappa^2.$$

Intuitively, $W_z^\kappa$ removes degenerated (non-active) directions of the decoder, ensuring that all eigenvalues of $G_{\text{Dec}}|_{W_z^\kappa}$ are bounded below by $\kappa$. This guarantees numerical stability for geometric computations requiring inversion of $G_{\text{Dec}}$. Note, that similarly to the full-rank case, $\hat{\text{Dec}}_{|W_z^\kappa} : (W_z^\kappa, G_{\text{Dec}}) \to (\hat{\text{Dec}}(W_z^\kappa), \langle \cdot, \cdot \rangle_{\mathbb{R}^{LA}})$ is an isometric diffeomorphism.

The explicit SVD-based construction of $W_z^\kappa$ is deferred to Appendix C. For stability near boundaries, we also use contracted domains $W_z^\kappa(\alpha) = \{z + \alpha(v - z) : v \in W_z^\kappa\}$ and $M_z^\kappa(\alpha) = (W_z^\kappa(\alpha), G_{\text{Dec}})$, with $\alpha \in (0, 1)$.

**On the usage of Euclidean metric in the ambient space**
Note that in the construction of the $\kappa$-stable manifolds, we endow the decoder output space, that is position-factorized peptide probabilities with the standard Euclidean metric. Although this choice may appear simplistic, it induces a mutationally uniform geometry that naturally penalizes non-local

insertions and deletions, which are more likely to induce substantial changes in peptide structure. In Appendix D we provide a formal derivation of these properties and discuss potential extensions to biologically informed ambient metrics.

## 2.3. SORBES - Second-Order Riemannian Brownian efficient Sampling

Having a stable Riemannian approximation $M_z^\kappa$ of $X$ in the vicinity of a point $z \in \mathcal{Z}$, we now describe how to explore it efficiently within the local neighbourhood of $z$. Our goal is to simulate a Riemannian Brownian motion for a time $T$ starting from $z$ that is a Riemannian equivalent of a local Gaussian perturbation $z + \epsilon$, $\epsilon \sim \mathcal{N}(0, T)$. To this end, we introduce the SORBES (Second-Order Riemannian Brownian efficient Sampling) procedure, described in Algorithm 1. SORBES improves the flat Gaussian noise by exploring only active local subspace $W_z^\kappa$, it is isotropic w.r.t. to the decoder geometry, and respects the local curvature of $M_z^\kappa$.

---

**Algorithm 1** SORBES

---

**Require:** $z \in \mathcal{Z}$, $\kappa \geq 0$, step size $\epsilon$, diffusion time $T$, $\alpha = 0.99$
 1: Initialize $z_0^\epsilon \leftarrow z$, stopped $\leftarrow$ False, $\sigma \leftarrow 0$
   ($\sigma$ tracks diffusion time)
 2: $W_z^\kappa, G_{\text{Dec}} \leftarrow M_z^\kappa$
 3: **for** $i = 1$ to $\lfloor \frac{T}{\epsilon^2} \rfloor$ **do**
 4:   $\overline{v} \in S_z^\kappa = \{u \in T_z M_z^\kappa : \langle u, u \rangle_z^{\text{Dec}} = 1\}$
     (Sample a unit tangent direction)
 5:   Set $v \leftarrow \sqrt{k_z^\kappa}\, \overline{v}$
 6:   **if** not stopped **then**
 7:

$$z_i^\epsilon = z_{i-1}^\epsilon + \underbrace{\epsilon v}_{\substack{\text{first-order} \\ \text{geodesic approximation}}} - \underbrace{\epsilon^2 \Gamma(z_{i-1}^\epsilon)[v,v]}_{\substack{\text{second-order} \\ \text{geodesic approximation}}},$$

     (Update - $\Gamma$ denotes the local Christoffel symbol for $M_z^\kappa$)
 8:   $\sigma \leftarrow \sigma + \epsilon^2$      (diffusion time update)
 9:   **if** $z_i^\epsilon \notin W_z^\kappa(\alpha)$ **then**
10:     stopped $\leftarrow$ True
11:   **end if**
12:   **else**
13:     $z_i^\epsilon \leftarrow z_{i-1}^\epsilon$      (absorbed state)
14:   **end if**
15: **end for**
   **return** $(z_i^\epsilon)_{0 \leq i \leq \lfloor \frac{T}{\epsilon^2} \rfloor}$, $\sigma$

---

Before introducing the key theoretical property of this algorithm, let's recall the crucial notation. For $A \subset M_z^\kappa$, $A^c$ is the complement of $A$ in $M_z^\kappa$. Let $d_{M_\kappa}$ be the geodesic distance on $M_z^\kappa$ w.r.t. to the pullback metric, and

$d_{M_z^\kappa}(x, A) = \inf_{y \in A} d_{M_z^\kappa}(x, y)$ for $x \in M_z^\kappa$ and $A \subset M_z^\kappa$. Let Ric be the Ricci curvature. Then the key theoretical property of the Algorithm 1 is summarized by:

**Theorem 2.1.** *Let $(Z_i^\epsilon)_{i \geq 0}$ be the sequence produced by Algorithm 1, for $M_z^\kappa(\alpha)$ with $\alpha \in (0, 1)$ and diffusion horizon $T > 0$, and define its continuous-time interpolation*

$$Z^\epsilon(t) := Z_{\lfloor \epsilon^{-2} t \rfloor}^\epsilon, \qquad t \geq 0.$$

*Let $R_\kappa^z = d_{M_z^\kappa}(z, (W_z^\kappa)^c)$, and suppose $L \geq 1$ satisfies*

$$\sup_{x \in M_z^\kappa(\alpha)} \text{Ric}_{M_z^\kappa}(x) \geq -L^2.$$

*Then for $T < \frac{(R_\kappa^z)^2}{4 k_z^\kappa L}$, as $\epsilon \to 0$, the process $Z^\epsilon$ converges in distribution to Riemannian Brownian motion stopped at the boundary of $M_z^\kappa(\alpha)$, with respect to the Skorokhod topology, on a set $C_{\kappa,z}^T \subset \Omega$ such that*

$$\mathbb{P}(C_{\kappa,z}^T) \geq 1 - \exp\left(-\frac{(R_\kappa^z)^2}{32T}\right).$$

For the proof, see Appendix E. Intuitively, in the small–step limit, our algorithm converges to Riemannian Brownian motion on $M_z^\kappa(\alpha)$, stopped at the boundary, with the deviation probability decaying exponentially in the inverse time horizon. Theorem 2.1 extends the main convergence result of Schwarz et al. (2023) to possibly non-compact manifolds. Importantly, Schwarz et al. (2023) showed that achieving this convergence requires a *second-order correction* term (capturing the effect of Christoffel symbols), rather than the commonly used naive first-order update. This motivates our use of the second-order scheme in Algorithm 1. In Appendix F (Algorithm 4), we describe SORBES-SE (*Stable/Efficient*), an implementation of SORBES that approximates the second-order correction using finite differences. It employs an adaptive step size $\epsilon$ that adjusts to the local curvature of the space, while still preserving the convergence guarantees.

## 2.4. MUTANG - Mutation Enumeration in Tangent Space

To further exploit the manifold structure of $M_z^\kappa$, modelling the neighborhood of a point $z \in \mathcal{Z}$, let us observe that the ambient tangent space $T_{\hat{\text{Dec}}(z)} \hat{\text{Dec}}(W_z^\kappa)$ identifies directions in peptide space along which the decoder output is the most sensitive. We interpret these directions as defining a *mutation space* for the decoded peptide p($z$), providing candidate amino-acid substitutions.

Formally, let $U^\kappa(z)$ (see Equation 8) denote an orthonormal basis of $T_{\hat{\text{Dec}}(z)} \hat{\text{Dec}}(W_z^\kappa)$ in the ambient space (Figure 2A–B), and let $u_j \in \mathbb{R}^{LA}$ be the $j$-th basis vector. We reshape

$u_j$ into matrix form

$$\Delta \operatorname{Dec}^{(j)}(z) = \texttt{reshape}(u_j, (L, A)) \in \mathbb{R}^{L \times A}. \quad (3)$$

Intuitively, each entry $\Delta \operatorname{Dec}^{(j)}(z)_{\ell,a}$ measures the first-order sensitivity of the probability assigned to amino acid $\mathcal{A}_a$ at position $\ell$, thereby suggesting a possible substitution. To extract candidate mutations, we introduce a sensitivity threshold $\theta_{\text{mut}} > 0$ and declare that

$$\left| \Delta \operatorname{Dec}^{(j)}(z)_{\ell,a} \right| \geq \theta_{\text{mut}} \quad \Rightarrow \quad \text{add mutation } \mathtt{p}(z)_\ell \to a, \quad (4)$$

where $\mathtt{p}(z)_\ell$ is the current residue at position $\ell$. Applying this rule across all $j = 1, \ldots, k_z^\kappa$ yields a *mutation pool*

$$\mathcal{P} \subseteq \{1, \ldots, L\} \times \mathcal{A}.$$

To enumerate candidate peptides, for each sequence position $\ell$ we define the set of admissible residues (Figure 2C) as

$$S_\ell = \{ \mathcal{A}_a \mid (\ell, a) \in \mathcal{P} \} \cup \{\mathtt{p}(z)_\ell\},$$

i.e., all suggested mutations together with the identity residue. The complete candidate set is then obtained as the Cartesian product (Figure 2D):

$$\mathcal{C}(\mathtt{p}(z)) = \prod_{\ell=1}^{L} S_\ell = \{ y \in \Sigma^L : y_\ell \in S_\ell \ \forall \ell \}. \quad (5)$$

The details of MUTANG are provided in Appendix G (Algorithm 5).

## 2.5. Local enumeration

We aim to densely populate the neighbourhood of a single prototype peptide with valid, diverse candidates. Starting from a seed peptide $\mathtt{p}$ with latent code $z_0$, we launch multiple Riemannian random walk trajectories using the SORBES algorithm (Section 2.3). At the start and after each step, the current latent state is decoded into a peptide and augmented with additional variants generated by MUTANG (Section 2.4). Additionally, to account for local dimension variability, we re-estimate the $\kappa$-stable submanifold $M_z^\kappa$ at each step of the random walk. The union of all decoded walk steps and tangent-space mutations yields a compact, high-quality *local candidate set* around $\mathtt{p}$. Despite leveraging second-order geometric information, LOCALENUMERATION remains computationally efficient: execution time of each step scales linearly with the latent dimension, peptide length, and decoder evaluation time. Algorithm 2 summarizes the procedure, with a detailed time and memory complexity as well as theoretical analysis provided in Appendix H.

## 2.6. LE-BO - Local Enumeration Bayesian Optimization

Finally, we integrate our LOCALENUMERATION algorithm into a Bayesian optimization (Garnett, 2023) framework for peptide design, which we term Local Enumeration Bayesian Optimization (LE-BO). Instead of performing costly optimization of the acquisition function in the latent space, which typically relies on Euclidean-distance kernels and ignores both the latent geometry and the discrete nature of peptides, we use surrogate Gaussian process models (Seeger, 2004), defined directly in the peptide space. The acquisition function is optimized by locally enumerating peptides in the vicinity of the most promising candidates and then selecting the peptide that maximizes the acquisition value. To further encourage exploration and increase the diversity of discovered candidates, we employ the ROBOT scheme (Maus et al., 2023), which promotes searching across a broader set of promising regions. Details of LE-BO are presented in Algorithm 3.

## 2.7. PoGS - Potential-minimizing Geodesic Search

Given two prototype peptides with latent vectors $z_a$ and $z_b$, we aim to generate *seeds*, i.e., *analogues that are jointly similar to both vectors and have high predicted activity*. To this end, we construct a discrete geodesic-like curve connecting $z_a$ and $z_b$, interpreted physically as a system with *kinetic energy* (geometric term) and an added *potential energy* (property term). This provides a natural tradeoff between similarity to both seeds and the desired molecular property.

Because the decoder Jacobian may have varying rank, we avoid intrinsic pullback computations and work in the ambient peptide-probability space $\mathbb{R}^{LA}$. For a sequence of latent waypoints $Z = \{z_0 = z_a, z_1, \ldots, z_N = z_b\}$, we define their decoded logits $X_k = \log(\hat{\operatorname{Dec}}(z_k))$, and approximate curve length using *chord distances* $\|X_{k+1} - X_k\|_2$. This extrinsic metric serves as a first-order surrogate for geodesic energy, bypassing costly Christoffel evaluations and remaining stable under rank variability.

We define the total energy of a discrete path $Z$ as

$$\mathcal{E}_{\lambda,\mu}(Z) = \underbrace{\sum_{k=0}^{N-1} \|X_{k+1} - X_k\|_2^2}_{\text{kinetic term: geometric similarity}} + \lambda \underbrace{\sum_{k=0}^{N} \Phi(X_k)}_{\text{potential term: property}}$$
$$+ \mu \underbrace{\sum_{k=0}^{N-1} \|z_{k+1} - z_k\|_2^2}_{\text{latent regularizer}},$$

$$(6)$$

where $\Phi$ is the property prediction (e.g. negative log MIC), $\lambda \geq 0$ balances geometry vs. property, and $\mu \geq 0$ regularizes latent jumps to discourage large chords in $\mathcal{Z}$. The first

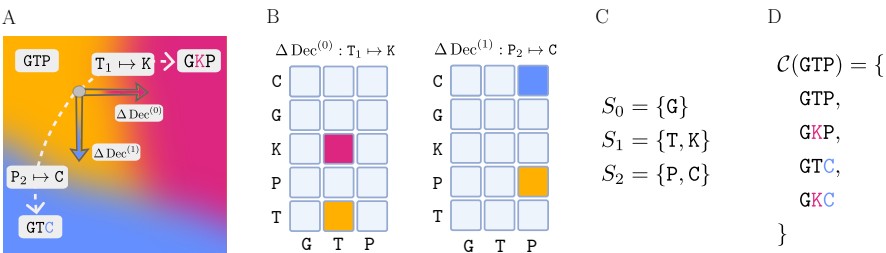

*Figure 2.* **Tangent space as mutation space and local enumeration. A)** For an example peptide GTP we consider two orthogonal peptide-space tangent directions $\Delta \mathrm{Dec}^{(j)}$ obtained from the SVD of the decoder Jacobian. Each direction suggests a specific substitution: $\mathtt{T}_1 \to \mathtt{K}$ and $\mathtt{P}_2 \to \mathtt{C}$. **B)** Each $\Delta \mathrm{Dec}^{(j)}$ is reshaped into an $L \times A$ map (rows: amino acids; columns: positions). **C)** Thresholded entries define per-position sets of admissible residues (identity always included). **D)** The candidate set is the Cartesian product $\mathcal{C}(\mathtt{GTP}) = \prod_\ell S_\ell$.

---

**Algorithm 2** LOCALENUMERATION

**Require:** seed peptide p, $\kappa_{\mathrm{SORBES}}, \kappa_{\mathrm{MUTANG}} \geq 0$, number of trajectories $M$, walk time $T_{\mathrm{walk}}$, step size $\epsilon$, mutation threshold $\theta_{\mathrm{mut}}$
1: Encode p to latent $z_0$; $\quad \mathcal{C} \leftarrow \{\mathtt{p}\}$
2: **for** $m = 1$ to $M$ **do**
3: $\quad z \leftarrow z_0; t \leftarrow 0; \mathcal{C} \leftarrow \mathcal{C} \cup \mathrm{MUTANG}(z, \kappa_{\mathrm{MUTANG}}, \theta_{\mathrm{mut}})$
4: $\quad$ **while** $t < T_{\mathrm{walk}}$ **do**
5: $\quad\quad ((\_, z), \sigma) \leftarrow \mathrm{SORBES\text{-}SE}(z, \kappa_{\mathrm{SORBES}}, \epsilon, \mathrm{STEP}_{\max}=1)$ $\qquad$ (single step of SORBES-SE)
6: $\quad\quad t \leftarrow t + \sigma^2; \quad \mathcal{C} \leftarrow \mathcal{C} \cup \{\mathtt{p}(z)\}$
7: $\quad\quad \mathcal{C} \leftarrow \mathcal{C} \cup \mathrm{MUTANG}(z, \kappa_{\mathrm{MUTANG}}, \theta_{\mathrm{mut}})$
8: $\quad$ **end while**
9: **end for**
$\quad$ **return** $\mathcal{C}$ $\qquad$ (local candidate set)

---

term corresponds to kinetic energy (favoring smooth, short ambient curves), the second to a potential energy that biases toward low $\Phi$, and the third acts as a stabilizer ensuring robustness of the chord points in the latent space.

To perform optimization and search, we initialize $Z$ by straight-line interpolation in latent space, and later optimize $\mathcal{E}_{\lambda,\mu}(Z)$ w.r.t. $Z$ using ADAM solver. During optimization, only the interior points $z_1, \ldots, z_{N-1}$ are updated. Given an optimized path $Z$, we decode each $z_i$ to a peptide $\mathtt{p}_i$ and remove consecutive duplicates, obtaining a *peptide path* $(\mathtt{p}'_0, \ldots, \mathtt{p}'_{N'})$ of length $N'$. We call a peptide $\mathtt{p}'_k$ a *seed* if its potential satisfies $\Phi(\mathtt{p}'_k) \leq \theta_{\mathrm{pot}}$ for a threshold $\theta_{\mathrm{pot}}$. A peptide $\mathtt{p}'_k$ is called a *well* if it is a seed and also a local minimum of the potential $\Phi$ along the peptide path. The detailed algorithm is presented in Appendix I (Algoritm 6).

## 3. Results

### 3.1. PoGS evaluation

To evaluate the POGS procedure, we applied it for the HydrAMP model (Szymczak et al., 2023), using average standardized MIC predictions against 3 *Escherichia coli* strains (*E. coli* ATCC11775, AIG221 and AIG222) of a APEX-derived transformer prediction (Wan et al., 2024)

(Appendix J) as the potential function. 300 prototype pairs $(z_a, z_b)$ were drawn from the Veltri dataset (Veltri et al., 2018), restricted to active peptides (average MIC $\leq 32\,\mu$g/ml against 3 *E. coli* strains) with edit distance between prototypes $\geq 10$.

For each pair $z_a$ and $z_b$, we compared three paths between $z_a$ and $z_b$: straight Euclidean interpolation, PoGS without potential, and full PoGS ; Section 2.7), measuring chord latent and ambient lengths, decoded peptide path lengths, and property-based counts of seeds and wells. PoGS hyperparametrs and metrics are described in Appendix I. PoGS achieved shorter ambient paths, and substantially more seeds and wells (Table 1), showing that property-aware potentials enrich trajectories for active candidates. In Appendix K we present additional analyses that further validate these findings, demonstrating that the observed gains are robust to the choice of ambient metric, set of prototypes and to the potential function.

### 3.2. LE-BO evaluation

We next evaluated our LE-BO algorithm on a black-box peptide optimization task with a budget of 1400 evaluations. Optimization was initialized from four seed peptides derived using PoGS for $\lambda = 0.01$ and $\mu = 0.1$ (parameters

*Table 1.* **PoGS results.** Comparison of straight interpolation, geodesic (no potential), and property-extended geodesic. Reported are latent length, ambient length, peptide path per length, and counts of seeds and wells.

| Method | Latent length | Ambient length | Peptide path length | Potential | Seeds | Wells |
|---|---|---|---|---|---|---|
| Straight interpolation | **6.43 ± 0.13** | 1601.56 ± 57.04 | 71.25 ± 6.12 | -501.00 ± 23.92 | 11.70 ± 5.54 | 2.50 ± 1.90 |
| PoGS w.o. potential ($\lambda = 0$) | 7.98 ± 0.18 | **1240.03 ± 40.72** | **66.68 ± 4.23** | -603.34 ± 15.12 | 18.90 ± 8.40 | 6.40 ± 3.20 |
| PoGS ($\lambda = 0.01$) | 9.02 ± 0.09 | 1432.55 ± 19.98 | 77.12 ± 6.12 | **-785.45 ± 34.12** | **22.70 ± 10.49** | **12.60 ± 4.45** |

---

**Algorithm 3** LE-BO

**Require:** ORACLE function to be optimized; seed $p_{seed}$; maximum budget $B_{max}$; trust region distance $d_{trust}$; number of ROBOT evaluations per iteration $k_{ROBOT}$; diversity threshold $d_{ROBOT}$; a surrogate Gaussian Process GP model with GP. acquistion function.

1: $p_{current} := p_{best} := p_{seed}$, $\mathcal{D} := \{p_{seed}\}$, $\mathcal{E} = \{(p_{seed}, \text{ORACLE}(p_{seed})\}$

2: **for** $iter := 1$ to $\lfloor B_{max}/k_{ROBOT} \rfloor$ **do**

3: $\quad \mathcal{D} := \mathcal{D} \cup \text{LOCALENUMERATION}(p_{current})$
$\quad$ (Explore the neighborhood of $p_{current}$)

4: $\quad$ GP := GP. fit($\mathcal{E}$) $\quad$ (Fit the surrogate GP model)

5: $\quad \mathcal{D}_{trust} := \{p \in \mathcal{D} \mid \text{Levenshtein}(p, p_{best}) \leq d_{trust}\}$
$\quad$ (Define the trust region)

6: $\quad$ **for** $i := 1$ to $k_{ROBOT}$ **do**

7: $\qquad p_{ROBOT}^i := \arg\max_{p \in \mathcal{D}_{trust}} \text{GP. acquistion}(p)$

8: $\qquad \mathcal{E} := \mathcal{E} \cup \{(p_{ROBOT}^i, \text{ORACLE}(p_{ROBOT}))\}$

9: $\qquad \mathcal{D}_{ROBOT} = \{p \in \mathcal{D} \mid \text{Levenshtein}(p, p_{ROBOT}^i) \leq d_{ROBOT}\}$

10: $\qquad \mathcal{D}_{trust} := \mathcal{D}_{trust} \setminus \mathcal{D}_{ROBOT}$
$\qquad$ (ROBOT diversity filtering)

11: $\quad$ **end for**

12: $\quad p_{current} := \arg\min_{1 \leq i \leq k_{ROBOT}} \text{ORACLE}(p_{ROBOT}^i)$

13: $\quad$ **if** $\text{ORACLE}(p_{current}) \geq \text{ORACLE}(p_{best})$ **then**

14: $\qquad p_{best} := p_{current}$

15: $\quad$ **end if**

16: **end for**

17: **return** $p_{best}$

---

described in Section 2.7): *KY14*, *KF16*, *KK16*, *FL14*, as well as two previously described AMPs (*mammuthusin-3*, *hydrodamin-2*) (Wan et al., 2024). As the optimized black-box function, we used the APEX MIC regressor (Wan et al., 2024), with the objective of minimization. We minimize the average $\log_2$ MIC across three *E. coli* strains, reporting the mean over the best results from 10 repeated optimization runs (Table 2). LE-BO hyperparameters are described in Appendix L.

We compared LE-BO against a diverse set of state-of-the-art methods, including generative models: HydrAMP (Szymczak et al., 2023) with different creativity hyperparameter $\tau$ and PepCVAE (Das et al., 2018); the diffusion-based model LaMBO-2 (Gruver et al., 2023); evolutionary strategies: Latent CMA-ES and Relaxed CMA-ES (Hansen, 2023), where

the former operates in the HydrAMP latent space and the latter optimizes directly on a continuous relaxation of one-hot sequence encodings, AdaLead (Sinai et al., 2020), PEX (Ren et al., 2022); GFlowNet-based approaches: GFN-AL (Jain et al., 2022) and GFN-AL-$\delta$CS (Kim et al., 2025); the reinforcement learning method DyNA-PPO (Angermueller et al., 2020); probabilistic methods: CbAS (Brookes et al., 2019) and Evolutionary BO (Sinai et al., 2020); the insertion-based Joker method (Porto et al., 2018); and greedy Random Mutation (González-Duque et al., 2024). Comparison to competitor latent BO methods SAASBO (Eriksson & Jankowiak, 2021), Hvarfner's Vanilla BO (Hvarfner et al., 2024), and LineBO (Kirschner et al., 2019) was limited by computational constraints: while LE-BO consistently completed within 2 hours, runs of these methods performing an equivalent 1400-step exploration in the HydrAMP latent space exceeded a one-week runtime budget (with SAASBO being the fastest among the latent BO baselines in our evaluation). Appendix M provides detailed time and memory profiling, including a direct comparison between LE-BO and SAASBO under matched experimental settings.

Additionally, we conducted an ablation study on LE-BO variants to isolate the effects of random walks and mutation enumeration. These variants modify the ENUMERATELO-CAL sub-procedure in Algorithm 2. The *Euclidean walk* variant replaces SORBES with a naive Euclidean random walk in the latent space. The *mutation-disabled* variant omits MUTANG. Finally, the *walk-disabled* variant uses only a single MUTANG without random walks.

As reported in Table 2, LE-BO achieved the best performance for five out of six prototypes, outperforming all baselines except for *FL14*, where it ranked second. The Ablated LE-BO variant with Euclidean walk and enabled mutations achieved the second-best performance on four out of six prototypes, underscoring the importance of mutation enumeration in the optimization process. These ablations further confirmed that both Riemannian random walks and mutations are essential for consistently achieving low MIC values. Additional analyses in Appendix N provide a geometric explanation for these trends. By adapting to the local decoder-induced geometry, Riemannian LE-BO identifies substantially larger and more diverse (Appendix O) local candidate sets for Bayesian optimization, particularly in regions with higher $\kappa$-stable dimensions, than Euclidean-based (Figure 8), explaining its stronger optimization perfor-

mance. Notably, these high-dimensional regions are associated with lower charge and aromaticity (Figure 9), and thus less favorable antimicrobial properties, where richer local enumeration increases the likelihood that such deficiencies can be corrected during optimization.

To further assess the reliability and generality of LE-BO, we conduct an extensive set of ablation and sensitivity analyses, summarized in Appendix P. These experiments evaluate the impact of key algorithmic components, including the stability parameters $\kappa_{\text{SORBES}}$ and $\kappa_{\text{MUTANG}}$, the removal of the ROBOT component, variation across different initial seeds, and the use of alternative oracle models. Since LE-BO employs only a single-step SORBES-SE walk, the contraction parameter $\alpha$ does not affect the resulting dynamics and is therefore omitted from the ablation study. Together, these analyses demonstrate that the performance gains of LE-BO are stable across algorithmic choices and experimental settings.

### 3.3. Wet-lab validation

To validate the computational predictions from PoGS and LE-BO, we conducted comprehensive *in vitro* antimicrobial testing of the generated peptides. A total of 29 novel peptides were experimentally evaluated: 4 seed peptides discovered through PoGS bi-prototype geodesics with property-aware potentials, and 25 analogs derived from these 4 seeds through LE-BO optimization for *E. coli* activity. These peptides were tested against a panel of 19 bacterial strains, including 8 multidrug-resistant (MDR) isolates (Table 14, Appendix Q), to assess both broad-spectrum activity and efficacy against clinically relevant resistant pathogens. We compared the success rate of PepCompass to previous methods that were also validated experimentally (HydrAMP (Szymczak et al., 2023), AMP-Diffusion (Torres et al., 2025), CLaSS (Das et al., 2021), Joker (Porto et al., 2018)). To this end, for each activity threshold, we computed the fraction of tested peptides that were active against at least one bacterial strain with this activity threshold. Because the prior methods were evaluated against different and partially non-overlapping bacterial panels, in Appendix Q (Figure 10) we additionally report success rates restricted to (A) the 11 strains shared between PepCompass and AMP-Diffusion and (B) *E. coli* strains shared across all four baselines.

As demonstrated in Figure 3, with unprecedented 100% success rate for the standard activity threshold of $32\mu g/ml$ and 82% rate at much more demanding threshold of $4\mu g/ml$, PepCompass achieved superior performance across various MIC thresholds when compared to previous methods. Moreover, these high success rates were maintained even when evaluated specifically against MDR bacterial strains (Figure 11, Table 15). The experimental results confirmed validity of the potential used in the optimization (Figure 12) as well

as the expected activity increase from prototypes to seeds to analogs for Gram-negative bacteria, directly validating our optimization strategy that targeted *E. coli* activity. Indeed, while the generated peptides showed some activity against Gram-positive bacterial strains (Figure 13), the clear enhancement from optimization was primarily observed against Gram-negative pathogens (Figure 14).

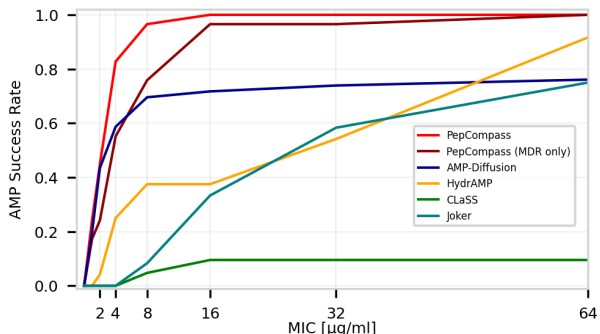

*Figure 3.* **Antimicrobial peptide success rates across MIC thresholds**. Success rate is defined as the fraction of generated peptides with MIC below the specified threshold against at least one tested strain. Results are based on experimental validation against 19 bacterial strains, including 8 MDR isolates.

## 4. Conclusions

By leveraging Riemannian latent geometry, interpretable tangent mutations, and potential-augmented geodesics, PepCompass enables efficient navigation and optimization within peptide space across global and local scales.

One potential limitation of the current implementation is the use of a Euclidean metric in the decoder output space. While seemingly simplistic, it induces a geometry that is mutationally uniform and structurally conservative to non-local insertions and deletions; see Appendix D for a rationale and potential extensions to more biologically informed metrics, which we plan to explore in future work. Another limitation arises from the construction of mutation candidates in MUTANG through a Cartesian product of local mutation sets. This formulation relies on a first-order approximation and implicitly assumes positional independence between mutation sites, which may not fully capture higher-order epistatic interactions present in biological sequences. Extending MUTANG to explicitly model multi-site dependencies therefore constitutes an important direction for future work. At the same time, the current formulation naturally subsumes single-site mutation strategies through the inclusion of the identity token at each position, and our experiments already demonstrate clear advantages over a purely single-site baseline (*Random Mutation*).

Already now, our computational experiments demonstrate superior performance over state-of-the-art baselines, and

*Table 2.* Minimal $\log_2$ MIC values achieved by each optimization method. Reported values are the mean and standard deviation over 10 runs. The top row shows the predicted $\log_2$ MIC of seeds before optimization. The first block reports baseline results, the second block shows ablations, and the last row presents our method. **Bold:** best overall value for a prototype. Underline: second-best.

| Method | | | KY14 | KF16 | KK16 | FL14 | mammuthusin-3 | hydrodamin-2 |
|---|---|---|---|---|---|---|---|---|
| $\log_2$ MIC value (seed) | | | 2.88 | 3.39 | 2.71 | 4.00 | 4.30 | 6.96 |
| GFN-AL | | | $2.73 \pm 0.27$ | $3.20 \pm 0.19$ | $2.63 \pm 0.17$ | $3.73 \pm 0.47$ | $4.30 \pm 0.00$ | $3.85 \pm 0.76$ |
| GFN-AL-$\delta$CS | | | $1.91 \pm 0.19$ | $1.74 \pm 0.20$ | $1.79 \pm 0.08$ | $2.06 \pm 0.42$ | $3.02 \pm 0.19$ | $1.77 \pm 0.41$ |
| PEX | | | $1.39 \pm 0.07$ | $1.48 \pm 0.09$ | $1.54 \pm 0.14$ | $1.27 \pm 0.27$ | $2.65 \pm 0.09$ | $1.14 \pm 0.11$ |
| Joker | | | $4.66 \pm 2.04$ | $3.98 \pm 1.52$ | $3.79 \pm 1.61$ | $4.22 \pm 1.08$ | $6.49 \pm 1.75$ | $6.28 \pm 0.33$ |
| Random Mutation | | | $1.23 \pm 0.26$ | $1.17 \pm 0.13$ | $0.97 \pm 0.40$ | $1.02 \pm 0.59$ | $2.12 \pm 0.80$ | $0.69 \pm 0.19$ |
| LaMBO-2 | | | $1.88 \pm 0.25$ | $2.11 \pm 0.20$ | $1.72 \pm 0.18$ | $1.76 \pm 0.32$ | $2.75 \pm 0.15$ | $2.17 \pm 0.35$ |
| Relaxed CMA-ES | | | $1.92 \pm 0.13$ | $1.83 \pm 0.28$ | $1.87 \pm 0.30$ | $1.93 \pm 0.40$ | $2.86 \pm 0.29$ | $1.66 \pm 0.34$ |
| Latent CMA-ES | | | $1.81 \pm 0.33$ | $2.07 \pm 0.56$ | $1.72 \pm 0.29$ | $1.72 \pm 0.20$ | $2.17 \pm 0.60$ | $1.87 \pm 0.44$ |
| CbAS | | | $2.88 \pm 0.00$ | $3.39 \pm 0.00$ | $2.71 \pm 0.00$ | $4.00 \pm 0.00$ | $4.30 \pm 0.00$ | $5.50 \pm 0.93$ |
| DyNAPPO | | | $1.45 \pm 0.36$ | $1.31 \pm 0.22$ | $1.34 \pm 0.20$ | $0.73 \pm 0.53$ | $2.42 \pm 0.54$ | $0.82 \pm 0.39$ |
| Evolutionary BO | | | $1.65 \pm 0.23$ | $1.45 \pm 0.42$ | $1.50 \pm 0.12$ | $1.70 \pm 0.19$ | $2.68 \pm 0.12$ | $1.28 \pm 0.33$ |
| AdaLead | | | $0.87 \pm 0.49$ | $1.01 \pm 0.28$ | $0.93 \pm 0.24$ | $\mathbf{0.51 \pm 0.38}$ | $2.30 \pm 0.28$ | $\underline{0.66 \pm 0.21}$ |
| PepCVAE | | | $3.66 \pm 0.00$ | $2.87 \pm 0.01$ | $3.14 \pm 0.11$ | $2.12 \pm 0.00$ | $4.30 \pm 0.00$ | $6.74 \pm 0.06$ |
| HydrAMP $\tau = 5.0$ | | | $2.35 \pm 0.08$ | $2.03 \pm 0.12$ | $1.88 \pm 0.10$ | $2.19 \pm 0.12$ | $3.19 \pm 0.27$ | $2.39 \pm 0.38$ |
| HydrAMP $\tau = 2.0$ | | | $2.60 \pm 0.03$ | $2.27 \pm 0.02$ | $2.11 \pm 0.11$ | $2.81 \pm 0.30$ | $3.99 \pm 0.01$ | $5.02 \pm 0.28$ |
| HydrAMP $\tau = 1.0$ | | | $2.86 \pm 0.01$ | $2.27 \pm 0.00$ | $2.35 \pm 0.00$ | $3.72 \pm 0.35$ | $4.27 \pm 0.10$ | $6.09 \pm 0.01$ |
| | Walk | Mutation | | | | | | |
| | Euclidean | – | $1.37 \pm 0.27$ | $1.33 \pm 0.23$ | $1.24 \pm 0.18$ | $1.29 \pm 0.17$ | $0.91 \pm 0.37$ | $1.16 \pm 0.24$ |
| | SORBES-SE | – | $1.37 \pm 0.22$ | $1.42 \pm 0.22$ | $1.12 \pm 0.35$ | $1.18 \pm 0.20$ | $1.07 \pm 0.50$ | $1.12 \pm 0.13$ |
| Ablated LE-BO | – | ✓ | $1.71 \pm 0.20$ | $1.46 \pm 0.40$ | $1.82 \pm 0.13$ | $1.24 \pm 0.13$ | $1.90 \pm 0.42$ | $1.12 \pm 0.20$ |
| | Euclidean | ✓ | $\underline{0.65 \pm 0.18}$ | $\underline{0.71 \pm 0.18}$ | $\underline{0.83 \pm 0.32}$ | $0.87 \pm 0.22$ | $\underline{0.78 \pm 0.45}$ | $0.80 \pm 0.18$ |
| LE-BO | SORBES-SE | ✓ | $\mathbf{0.50 \pm 0.24}$ | $\mathbf{0.60 \pm 0.29}$ | $\mathbf{0.50 \pm 0.14}$ | $\underline{0.60 \pm 0.22}$ | $\mathbf{0.50 \pm 0.38}$ | $\mathbf{0.58 \pm 0.34}$ |

wet-lab validation confirms unprecedented success rates, with all tested peptides showing *in vitro* activity, including against multidrug-resistant pathogens. These results establish geometry-aware exploration as a powerful paradigm for controlled generative design in vast biological spaces.

## Impact Statement

This work advances the field of machine learning by developing a principled, geometry-aware framework (PepCompass) for navigating high-dimensional generative latent spaces.

Antimicrobial resistance (AMR) is a pressing public health challenge worldwide, leading to increased morbidity, mortality, and healthcare costs. Machine learning that accelerates *d*e novo design of effective antimicrobial agents can contribute positively by expanding the repertoire of candidate therapeutics and potentially reducing time and cost in early drug discovery. By facilitating systematic exploration of peptide design spaces, this work may help identify novel active compounds that would be difficult or costly to find through traditional methods alone.

At the same time, techniques that enhance biological sequence design could theoretically be repurposed, intentionally or unintentionally, to optimize sequences with harmful biological activity. While our current work focuses on antimicrobial peptides against pathogenic bacteria, any general framework for biological sequence optimization warrants careful consideration of misuse scenarios.

### Acknowledgments

Cesar de la Fuente-Nunez holds a Presidential Professorship at the University of Pennsylvania. Research reported in this publication was supported by NIH R35GM138201 and DTRA HDTRA1-21-1-0014. This project has received funding from the European Research Council (ERC) under the European Funding Union's Horizon 2020 research and innovation programme (grant agreement No 810115 – DOG-AMP). Adam Bielecki and Jurand Prądzyński were supported by "Excellence Initiative – Research University" Programme at the University of Warsaw.

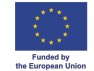 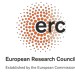

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

## A. Related work

**Riemannian latent geometry.**  Deep generative models can be endowed with Riemannian structure (Flaherty & do Carmo, 2013) by pulling back the ambient metric through the decoder Jacobian (Shao et al., 2017; Arvanitidis et al., 2021b;a). This allows distances and geodesics to reflect data geometry rather than Euclidean latent coordinates. However, most approaches assume a single smooth manifold of fixed dimension. Evidence from both theory and experiments shows that data often lie on unions of manifolds with varying intrinsic dimension or CW-complex structures (?Brown et al., 2023; Wang & Wang, 2024). In practice, existing methods either restrict the latent to very low dimensions ($\leq 8$) or inflate the metric with variance terms to ensure full rank (Arvanitidis et al., 2021b; Detlefsen et al., 2022), but these ignore extrinsic geometry. Our model instead *decomposes the latent into Riemannian submanifolds of varying dimensions*, enabling principled geometry across heterogeneous regions.

**Brownian motion and random walks.**  Latent Brownian motion has been used as a prior for VAEs (Kalatzis et al., 2020) and for Riemannian score-based modeling (De Bortoli et al., 2022). But these approaches rely on first-order updates. Convergence results for geodesic random walks show that correct Riemannian and sub-Riemannian Brownian motion requires second-order approximations (Schwarz et al., 2023; Herrmann et al., 2023). Our method explicitly incorporates this requirement, yielding diffusion-consistent walks where previous methods diverge.

**Tangent spaces and interpretability.**  Tangent-space analysis has mostly been applied in vision, where interpretable latent directions are discovered in GANs or diffusion models via Jacobian or eigen decompositions (Shen et al., 2020; Dombrowski et al., 2021; Zhu et al., 2021; Park et al., 2023). Frames induced by augmentations provide another lens on local tangent geometry (Kvinge et al., 2024). We are the first to provide an *interpretable tangent space in peptide sequence models*, where tangent vectors correspond directly to biologically meaningful mutations.

**Geodesics and potentials.**  Geodesics are widely used for interpolation and counterfactual reasoning in latent space (Pegios et al., 2024; Kim et al., 2024). Yet these are typically free geodesics. We extend the concept with *potentials*, leveraging the Jacobi metric (Gibbons, 2015) so that peptide traversals account for both geometry and biochemical preferences.

**Applications in molecules and proteins.**  Geometry-aware latent models have been used for chemical-space exploration (Du et al., 2022; Wei et al., 2024), and protein sequence modeling (Moreta, L., Rønning, O., Al-Sibahi, A., Hein, J., Theobald, D., Hamelryck, T., 2022; Detlefsen et al., 2022). Our approach complements these by combining: (i) varying-dimension latent decomposition, (ii) second-order consistent Brownian walks, (iii) interpretable tangent spaces via mutations, and (iv) potential-augmented geodesics tailored to peptide design.

## B. $\kappa$-stable dimension of the PepCVAE (Das et al., 2018) and HydrAMP (Szymczak et al., 2023) models

To quantify how the $\kappa$-stable dimension varies across the latent space, we sampled $1,000$ points from the HydrAMP (Szymczak et al., 2023) training set and computed their $\kappa$-stable dimensions under both the PepCVAE (Das et al., 2018) and HydrAMP (Szymczak et al., 2023) models. We set $\kappa = 10^{-8}$, corresponding to the precision of the `float32` format commonly used in neural network computations. This choice ensures that no eigenvalue of the inverse metric tensor exceeds $10^8$, thereby avoiding numerical instabilities.

As shown in Figure 4A, the $\kappa$-stable dimension was always strictly below the latent dimensionality (64) of both models. This indicates that the effective local dimensionalities of the peptide spaces are substantially smaller than the nominal latent dimension. Furthermore, when comparing peptide length (Figure 4B) to $\kappa$-stable dimension (Figure 4C), we observe a clear positive correlation: longer peptides systematically yield higher $\kappa$-stable dimensions. Intuitively, this suggests that longer sequences admit more locally meaningful perturbations, which naturally translate into a richer set of candidate substitutions. This observation further justifies our MUTANG strategy (§2.4), as it allocates a larger and more diverse mutation pool precisely where biological sequence length provides greater combinatorial flexibility.

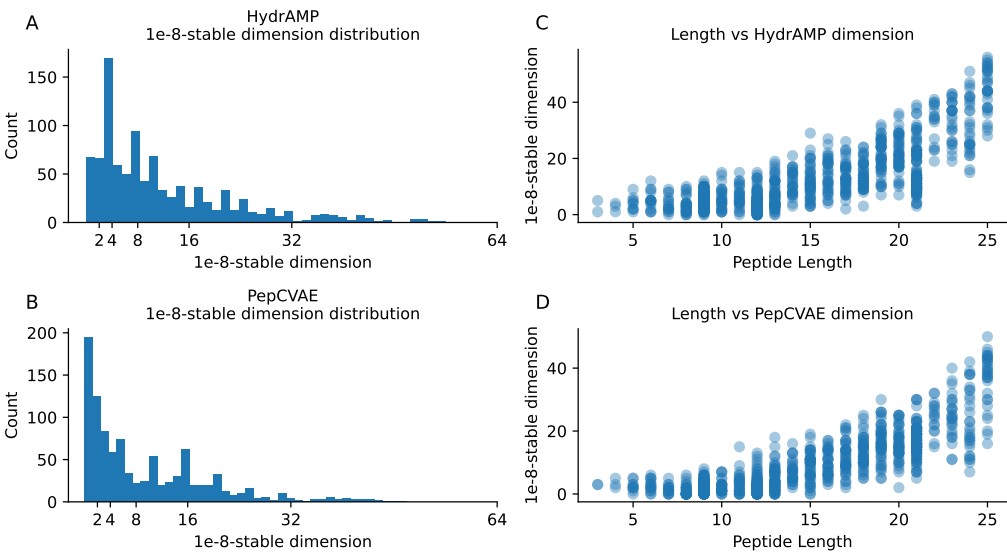

*Figure 4.* **Stable rank and peptide statistics. A–B)** Distributions of the $\kappa$-stable dimension ($\kappa = 10^{-8}$) for HydrAMP (A) and PepCVAE (B) across sampled peptides. **C–D)** Scatter plots of peptide length versus $\kappa$-stable dimension for HydrAMP (C) and PepCVAE (D), revealing a clear positive correlation: longer peptides tend to yield higher stable dimensions.

## C. Construction of $\kappa$-Stable Manifolds

In this section we will introduce a construction of $\kappa$-stable Riemannian submanifolds, namely

$$\mathbb{M}^{\kappa} = \{M_z^{\kappa} : z \in \mathcal{Z}\}, \qquad M_z^{\kappa} = (W_z^{\kappa}, G_{\mathrm{Dec}}),$$

where each $W_z^{\kappa} \ni z$ is an open affine submanifold of *maximal dimension* (denoted $k_z^{\kappa}$) through $z$, such that the pullback metric $G_{\mathrm{Dec}}$ restricted to $W_z^{\kappa}$, denoted $G_{\mathrm{Dec}}|_{W_z^{\kappa}}$, has full rank and satisfies the $\kappa$-stability condition

$$\inf_{\substack{v \in T_z W_z^{\kappa} \\ \langle v,v \rangle_{\mathbb{R}^d} = 1}} \langle v, v \rangle_z^{\mathrm{Dec}} > \kappa^2.$$

For this, we will use the truncated-SVD of a flattened decoder Jacobian $J_{\hat{Dec}}$. Let

$$J_{\hat{\mathrm{Dec}}}(z) = U(z)\,\Sigma(z)\,V(z)^{\top}$$

be the thin SVD of the decoder Jacobian, with singular values $\sigma_0(z) \geq \sigma_1(z) \geq \cdots \geq 0$, and $\Sigma(z) = \mathrm{diag}\left((\sigma_0, \sigma_1, \ldots, \sigma_{d-1})\right)$. Now let us note that:

$$G_{\mathrm{Dec}}(z) = J_{\hat{\mathrm{Dec}}}(z)^{\top} J_{\hat{\mathrm{Dec}}}(z) = V(z)\Sigma^2(z)V(z)^{\top}, \tag{7}$$

and define the *$\kappa$-stable dimension*:

$$k_z^{\kappa} = \#\{\, i : \sigma_i(z)^2 > \kappa \,\}.$$

Truncating SVD-decomposition to first $k_z^{\kappa}$ singular values gives

$$U^{\kappa}(z) \in \mathbb{R}^{(LA) \times k_z^{\kappa}}, \quad \Sigma_z^{\kappa} \in \mathbb{R}^{k_z^{\kappa} \times k_z^{\kappa}}, \quad V^{\kappa}(z) \in \mathbb{R}^{d \times k_z^{\kappa}}, \tag{8}$$

with truncated Jacobian:

$$J_{\hat{\mathrm{Dec}}}^{\kappa} = U(z)^{\kappa}\Sigma(z)^{\kappa}V(z)^{\kappa} \in \mathbb{R}^{LA \times d}.$$

We then define the affine subspace (together with its parametrization)

$$\mathcal{V}_z^{\kappa} = \{\phi_z^{\kappa}(x) : x \in \mathbb{R}^{k_z^{\kappa}}\}, \qquad \phi_z^{\kappa}(x) = z + V^{\kappa}(z)x.$$

Restricting the decoder to $\mathcal{V}_z^\kappa$ gives $\mathrm{Dec}_z^\kappa = \hat{\mathrm{Dec}} \circ \phi_z^\kappa$, with Jacobian

$$J_{\mathrm{Dec}_z^\kappa}(0) = U^\kappa(z)\Sigma^\kappa(z), \tag{9}$$

which has full column rank $k_z^\kappa$. By the inverse function theorem, there exists a neighborhood $\hat{W}_z^\kappa = B(0, r_z^\kappa)$ such that $\mathrm{Dec}_z^\kappa(\hat{W}_z^\kappa)$ is a smooth $k_z^\kappa$-dimensional manifold. Its pullback metric is

$$G_{\mathrm{Dec}_z^\kappa}(0) = \left(\Sigma^\kappa(z)\right)^2, \tag{10}$$

with eigenvalues $\{\sigma_i(z)^2 : \sigma_i(z)^2 > \kappa\}$, all $\geq \kappa$.

Finally, set $W_z^\kappa = \phi_z^\kappa(\hat{W}_z^\kappa)$. Then

$$M_z^\kappa = (W_z^\kappa, G_{\mathrm{Dec}}),$$

and $(\phi_z^\kappa)^{-1}$ is a diffeomorphic isometry between $(W_z^\kappa, G_{\mathrm{Dec}})$ and $(\hat{W}_z^\kappa, G_{\mathrm{Dec}_z^\kappa})$. From this and Eq. 10 it follows that $M_\kappa^z$ satisfies the $\kappa$-stability condition.

The maximality of $k_z^\kappa$ follows from the fact that if there existed an affine subspace $\bar{W}_\kappa^z$ with $\dim(\bar{W}_\kappa^z) > k_z^\kappa$ such that $G_{\mathrm{Dec}}|_{\bar{W}_z^\kappa}$ has full rank and satisfies the $\kappa$-stability condition, then we could define

$$\bar{W}_z^\kappa \cap (W_z^\kappa)^\perp = \left\{w \in \bar{W}_\kappa^z : \forall v \in W_z^\kappa, \ \langle v, w \rangle^{\mathrm{Euc}} = 0\right\} \subset \bar{W}_z^\kappa,$$

as the subspace of $\bar{W}_z^\kappa$ orthogonal to $W_z^\kappa$. Since $\dim(\bar{W}_\kappa^z) > \dim(W_\kappa^z)$, it follows that $\dim\left(\bar{W}_z^\kappa \cap (W_z^\kappa)^\perp\right) > 0$. By construction

$$\bar{W}_z^\kappa \cap (W_z^\kappa)^\perp \subset \mathrm{span}\{V(z)_{:,k_z^\kappa}, \dots, V(z)_{:,d}\}.$$

what implies that for all $v \in \bar{W}_z^\kappa \cap (W_z^\kappa)^\perp$ it holds that

$$v = a_{k_z^\kappa}V(z)_{:,k_z^\kappa} + \cdots + a_{d-1}V(z)_{:,d-1},$$

for some $a_{k_z^\kappa}, \dots, a_{d-1} \in \mathbb{R}$. Now take $v \in \bar{W}_z^\kappa \cap (W_z^\kappa)^\perp$ such that $\langle v, v \rangle^{\mathrm{Euc}} = 1$. Equation 7 then implies

$$\langle v, v \rangle_z^{\mathrm{Dec}} = a_{k_z^\kappa}^2 \sigma_{k_z^\kappa}^2 \|V(z)_{:,k_z^\kappa}\|^2 + \cdots + a_{d-1}^2 \sigma_{d-1}^2 \|V(z)_{:,d-1}\|^2 \tag{11}$$

$$\leq \kappa^2 \left(a_{k_z^\kappa}^2 \|V(z)_{:,k_z^\kappa}\|^2 + \cdots + a_{d-1}^2 \|V(z)_{:,d-1}\|^2\right) \tag{12}$$

$$= \kappa^2, \tag{13}$$

which contradicts the assumption that $\bar{V}_z^\kappa$ satisfies the $\kappa$-stability condition.

**Note.** This construction cannot be replaced by a direct Frobenius theorem argument, since $k_z^\kappa = \ell$ at a point does not imply that $k^\kappa$ is constant in a neighborhood (see Appendix C.1).

### C.1. On the local instability of $\kappa$-stable dimension

Recall that $k_z^\kappa = \#\{i : \sigma_i(z) > \sqrt{\kappa}\}$ counts the number of singular values of $J_{\mathrm{Dec}}(z)$ exceeding the threshold $\sqrt{\kappa}$. While $k_z^\kappa$ is well defined at every point $z$, it need not be locally constant. In particular, singular values of $J_{\mathrm{Dec}}(z)$ depend continuously on $z$, but they can cross the threshold $\sqrt{\kappa}$ arbitrarily close to a given point. Hence, even if $k_z^\kappa = \ell$ at some $z$, there may exist nearby points $z'$ with $k_{z'}^\kappa > \ell$ (see Figure 5).

This observation prevents a direct application of the Frobenius or constant rank theorem, which require a rank function that is constant in a neighborhood. Our construction in Section 2.1 circumvents this issue by working with an open set $W_z^\kappa$ around $z$ on which the rank remains constant, thereby ensuring that both $M_z^\kappa$ and $\mathrm{Dec}(W_z^\kappa)$ are smooth $k_z^\kappa$-dimensional submanifolds.

## D. Mutational Interpretation of the Ambient Euclidean Metric

In this section we elaborate on the interpretation of the Euclidean metric used in the decoder output space $\hat{X} \subset \mathbb{R}^{LA}$ and its implications for the geometry of peptide modifications, as well as on possible metric extensions.

*Figure 5.* **Example of a local non-stability of a $\kappa$-stable dimension.** A function $f(x) = x^3 + \kappa x$ has a stable rank $k^\kappa$ equal to 1 everywhere except of 0. So in every neighbourhood of 0, a stable rank is different than 0, thus preventing the application of a Frobenious theorem.

### D.1. Euclidean mutation geometry

A discrete peptide sequence p of length $L$ over an alphabet $\mathcal{A}$ of size $A$ is represented as a vector $\text{oh}(\text{p}) \in \{0, 1\}^{LA} \subset \mathbb{R}^{LA}$ obtained by concatenating $L$ one-hot vectors,

$$\text{oh}(\text{p}) = (e_{a_1}, \dots, e_{a_L}),$$

where $e_{a_\ell} \in \mathbb{R}^A$ denotes the canonical basis vector corresponding to amino acid $a_\ell$ at position $\ell$.

Let $\text{mut}_{a \to b, \ell}$ denote the substitution of amino acid $a$ by $b$ at position $\ell$. For any peptide p with $a_\ell = a$,

$$\text{oh}(\text{mut}_{a \to b, \ell}(\text{p})) - \text{oh}(\text{p}) = \psi_{a \to b, \ell},$$

where

$$\psi_{a \to b, \ell} = (0, \dots, e_b - e_a, \dots, 0) \in \mathbb{R}^{LA}.$$

For two substitutions $\text{mut}_{a \to b, \ell_1}$ and $\text{mut}_{c \to d, \ell_2}$,

$$\langle \psi_{a \to b, \ell_1}, \psi_{c \to d, \ell_2} \rangle = \mathbf{1}_{\{\ell_1 = \ell_2\}} \big( \mathbf{1}_{\{a=c\}} + \mathbf{1}_{\{b=d\}} - \mathbf{1}_{\{a=d\}} - \mathbf{1}_{\{b=c\}} \big).$$

Consequently,

$$\|\psi_{a \to b, \ell}\|_2^2 = 2,$$

substitutions acting on different positions are orthogonal, and substitutions at the same position have constant pairwise cosine similarity $1/2$ (corresponding to an angle of $30°$). Thus, the Euclidean metric induces a uniform and isotropic geometry over substitutional mutations.

This implies that using a Euclidean metric in the ambient amino-acid probability space can be interpreted as imposing a non-informative prior over mutation directions.

### D.2. Euclidean insertion and deletion geometry

Let $\text{ins}_{a, \ell}$ and $\text{del}_\ell$ denote insertion of amino acid $a$ at position $\ell$ and deletion at position $\ell$, respectively. Terminal insertions and deletions can be treated as substitutions involving a padding symbol $\text{pad}$ and therefore have squared norm 2, and are orthogonal to other substitutions.

For internal insertions and deletions, shifting of downstream residues induces a perturbation, whose squared Euclidean norm satisfies

$$\|\text{oh}(\text{ins}_{a, \ell}(\text{p})) - \text{oh}(\text{p})\|_2^2 = 2 \Big( \mathbf{1}_{\{\text{p}_\ell = a\}} + \sum_{i=\ell}^{L-1} \mathbf{1}_{\{\text{p}_{i+1} \neq \text{p}_i\}} \Big) \leq 2(L - \ell + 1),$$

and

$$\|\mathrm{oh}(\mathtt{del}_\ell(\mathbf{p})) - \mathrm{oh}(\mathbf{p})\|_2^2 = 2\Big(1 + \sum_{i=\ell}^{L-1} \mathbf{1}_{\{\mathbf{p}_{i+1} \neq \mathbf{p}_i\}}\Big) \leq 2(L - \ell + 1).$$

The potentially large displacement reflects the non-local nature of internal insertions and deletions, and appropriately penalizes mutations which are likely to induce global structural rearrangements, such as register shifts or changes in secondary structure.

### D.3. Extension to biologically meaningful ambient metrics by mutation kernels

Let
$$\mathtt{Mut} = \{\mathtt{mut}_{a \to b, \ell} : a, b \in \mathcal{A}, \ \ell \in \{1, \ldots, L\}\}$$
be the set of substitution operators, and consider a positive-definite mutation kernel
$$\mathcal{K}_{\mathtt{Mut}} : \mathtt{Mut} \times \mathtt{Mut} \to \mathbb{R}.$$

Here, diagonal terms of $\mathcal{K}_{\mathtt{Mut}}$ control mutation magnitudes, while off-diagonal terms encode similarity between distinct mutations, enabling incorporation of biochemical substitution preferences and positional coupling into the ambient geometry.

Define
$$V_{\mathtt{Mut}} = \mathrm{span}\{\psi_{a \to b, \ell} : \mathtt{mut}_{a \to b, \ell} \in \mathtt{Mut}\}.$$
For $v_1 = \sum_i c_i \psi_i$ and $v_2 = \sum_j d_j \psi_j$, define
$$\mathcal{K}_{\mathcal{K}_{\mathtt{Mut}}}^{V_{\mathtt{Mut}}}(v_1, v_2) = \sum_{i,j} c_i d_j \, \mathcal{K}_{\mathtt{Mut}}(\mathtt{mut}_i, \mathtt{mut}_j).$$

Every $w \in \mathbb{R}^{LA}$ decomposes uniquely as $w = w^{V_{\mathtt{Mut}}} + w^{V_{\mathtt{Mut}}^\perp}$ under the standard Euclidean inner product. Define the ambient kernel
$$\mathcal{K}_{\mathcal{K}_{\mathtt{Mut}}}^{\mathbb{R}^{LA}}(w_1, w_2) = \mathcal{K}_{\mathcal{K}_{\mathtt{Mut}}}^{V_{\mathtt{Mut}}}(w_1^{V_{\mathtt{Mut}}}, w_2^{V_{\mathtt{Mut}}}) + \langle w_1^{V_{\mathtt{Mut}}^\perp}, w_2^{V_{\mathtt{Mut}}^\perp}\rangle.$$

This construction transfers biologically meaningful similarity between mutations to a Riemannian metric on the ambient space. We plan to explore such metrics in future work.

**Extension of $\kappa$-stable manifold construction to kernel-based ambient metrics**   Let $e_i \in \mathbb{R}^{LA}$, $i = 1, \ldots, LA$, denote the canonical basis vectors, and define the Gram matrix
$$G_{\mathcal{K}}[i, j] = \mathcal{K}_{\mathcal{K}_{\mathtt{Mut}}}^{\mathbb{R}^{LA}}(e_i, e_j).$$

Let
$$G_{\mathcal{K}} = U_{\mathcal{K}} \Sigma_{\mathcal{K}} U_{\mathcal{K}}^\top$$
be its eigen decomposition. Define a modified decoder
$$\hat{\mathrm{Dec}}_{\mathcal{K}}(z) = \Sigma_{\mathcal{K}}^{1/2} U_{\mathcal{K}}^\top \hat{\mathrm{Dec}}(z).$$

Then
$$J_{\hat{\mathrm{Dec}}_{\mathcal{K}}}(z) = \Sigma_{\mathcal{K}}^{1/2} U_{\mathcal{K}}^\top J_{\hat{\mathrm{Dec}}}(z),$$
and the induced pullback metric satisfies
$$J_{\hat{\mathrm{Dec}}_{\mathcal{K}}}(z)^\top J_{\hat{\mathrm{Dec}}_{\mathcal{K}}}(z) = J_{\hat{\mathrm{Dec}}}(z)^\top G_{\mathcal{K}} J_{\hat{\mathrm{Dec}}}(z).$$

Thus, equipping the ambient space with the Mahalanobis inner product induced by $G_{\mathcal{K}}$ and pulling it back through $\hat{\mathrm{Dec}}$ is equivalent to equipping the ambient space with the standard Euclidean inner product and pulling it back through $\hat{\mathrm{Dec}}_{\mathcal{K}}$ (Flaherty & do Carmo, 2013). In other words, both models induce the same Riemannian metric on latent space. Consequently, the $\kappa$-stable manifold construction from Subsection 2.2 can be applied for $\hat{\mathrm{Dec}}_{\mathcal{K}}$, and the same argument extends to any fixed Mahalanobis-type ambient metric on $\mathbb{R}^{LA}$.

# E. Proof of Theorem 2.1

In this section we prove the main theorem of the paper. As preparation, we first recall the definition of Riemannian Brownian motion, starting with the compact case.

**Riemannian Brownian Motion (compact case).** Let $(M, g)$ be a smooth, compact, connected, $d$-dimensional Riemannian manifold with Riemannian metric $g$. A *Riemannian Brownian motion* on $M$ is a continuous stochastic process

$$B = \{B_t\}_{t \geq 0}$$

defined on a filtered probability space $(\Omega, \mathcal{F}, \{\mathcal{F}_t\}_{t \geq 0}, \mathbb{P})$, satisfying:

1. $B_0 = x \in M'$ almost surely, for some fixed starting point $x \in M$.

2. The sample paths $t \mapsto B_t$ are almost surely continuous and adapted to the filtration $\{\mathcal{F}_t\}$.

3. For every smooth function $f \in C^\infty(M)$, the process

$$f(B_t) - f(B_0) - \tfrac{1}{2} \int_0^t (\Delta_g f)(B_s) \, ds$$

   is a real-valued local martingale, where $\Delta_g$ denotes the Laplace–Beltrami operator associated with $g$.

4. The generator of $B_t$ is $\frac{1}{2}\Delta_g$, i.e.

$$\lim_{t \to 0} \frac{\mathbb{E}[f(B_t)] - f(x)}{t} = \tfrac{1}{2}(\Delta_g f)(x), \quad \forall f \in C^\infty(M).$$

**Extension to the non-compact case: smooth spherical-cap compactification.** Our manifolds of interest, $M_z^\kappa(\alpha)$, $\alpha \in (0, 1)$, $z \in \mathcal{Z}$, are open subsets of $M_z^\kappa$ and therefore non-compact. To define Brownian motion in this setting, one needs to control the behaviour of paths near the boundary. Classical approaches include: (i) compactification (Wang, 2011), (ii) stopping the process at the boundary (Hsu, 2002), or (iii) reflecting it (Du & Hsu, 2023). In our work we adopt a compactification strategy via a smooth spherical cap (see Figure 6), followed by stopping on the boundary of a natural embedding of $M_z^\kappa(\alpha)$.

Concretely, let $k_z^\kappa$ be the $\kappa$-stable dimension and consider the unit sphere

$$S_1^{k_z^\kappa} \subset \mathbb{R}^{k_z^\kappa + 1}.$$

Take an atlas of this sphere consisting of two charts $(U_1, \psi_1)$, $(U_2, \psi_2)$ such that $U_1, U_2 \subset \mathbb{R}^{k_z^\kappa}$ $\psi_1(U_1) \cup \psi_2(U_2) = S_1^{k_z^\kappa}$, with $\alpha \bar{W}_z^\kappa \subset U_1$ ($\alpha \bar{W}_z^\kappa = \{\alpha v : v \in \bar{W}_z^\kappa\}$) and

$$\psi_1 \circ \psi_2^{-1}(U_2) \cap \bar{W}_z^\kappa = \emptyset.$$

Let $G_{\psi_i}$ be the pullback metric on $U_i$ induced by $\psi_i$.

Choose a smooth bump function $b \in C^\infty$ such that

$$b \equiv 1 \quad \text{on } (\tfrac{2}{3}\alpha + \tfrac{1}{3})\bar{W}_z^\kappa, \qquad b \equiv 0 \quad \text{on } (\tfrac{1}{3}\alpha + \tfrac{2}{3})\bar{W}_z^\kappa.$$

We then define the compactified manifold $\overline{M_z^\kappa}(\alpha)$ with charts $\{(U_1, \psi_1), (U_2, \psi_2)\}$ and Riemannian metric on $U_1$ given by

$$G = b \, G_{\mathrm{Dec}_z^\kappa} + (1 - b) \, G_{\psi_1},$$

where

$$\overline{\mathrm{Dec}_z^\kappa}(x) = \begin{cases} \mathrm{Dec}_z^\kappa(x), & x \in \alpha \bar{W}_z^\kappa, \\ 0, & \text{otherwise.} \end{cases}$$

By construction, there is a natural isometric embedding

$$M_z^\kappa(\alpha) \hookrightarrow \overline{M_z^\kappa}(\alpha).$$

This compactification allows us to invoke convergence results for Brownian motion on compact manifolds, while ensuring that in the region of interest the geometry coincides with that of the original $\kappa$-stable manifold.

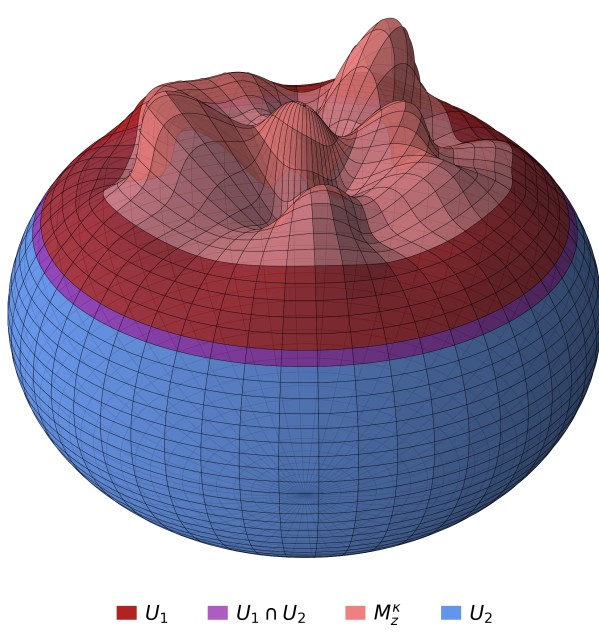

$U_1$    $U_1 \cap U_2$    $M_z^\kappa$    $U_2$

*Figure 6.* **Sphere-cap manifold compactification** $\overline{M_z^\kappa}$. Illustration of how the $M_z^\kappa$ manifold is compactified by capping off the spherical domain with the appropriate boundary conditions.

*Theorem* (Main Theorem). Let $(Z_i^\epsilon)_{i \geq 0}$ be the sequence produced by Algorithm 1, for $M_z^\kappa(\alpha)$ with $\alpha \in (0, 1)$ and diffusion horizon $T > 0$, and define its continuous-time interpolation

$$Z^\epsilon(t) := Z_{\lfloor \epsilon^{-2} t \rfloor}^\epsilon, \qquad t \geq 0.$$

Let $R_\kappa^z = d_{M_z^\kappa}(z, \alpha W_z^\kappa)^c)$, and suppose $L \geq 1$ satisfies

$$\sup_{x \in M_z^\kappa(\alpha)} \mathrm{Ric}_{M_z^\kappa}(x) \geq -L^2.$$

Then for $T < \frac{(R_z^\kappa)^2}{4 k_z^\kappa L}$, as $\epsilon \to 0$, the process $X^\epsilon$ converges in distribution to Riemannian Brownian motion stopped at the boundary of $M_z^\kappa(\alpha)$, with respect to the Skorokhod topology, on a set $C_{\kappa, z}^T \subset \Omega$ such that

$$\mathbb{P}(C_{\kappa, z}^T) \geq 1 - \exp\left(-\frac{(R_z^\kappa)^2}{32T}\right).$$

*Proof.* Let $\overline{M_z^\kappa(\alpha)}$ be the spherical-cap compactification of $M_z^\kappa(\alpha)$. Schwarz et al. (2023) showed that the non-stopped version of Algorithm 1, extended to $\overline{M_z^\kappa(\alpha)}$, produces a process $\overline{Z^\epsilon}$ that converges to $B$ in the Skorokhod topology on $\overline{M_z^\kappa(\alpha)}$.

For a closed set $A$ define the exit time

$$T_A^X = \inf\{t \in [0, T] : X(t) \notin A\} \wedge T$$

Both $T_{\alpha W_z^\kappa}^{\overline{X^\epsilon}}$ and $T_{\alpha W_z^\kappa}^B$ are valid stopping times (by right-continuity and because $(W_z^\kappa)^c$ is closed), and (from the construction of the Algorithm 1:

$$Z^\epsilon = \overline{Z^\epsilon}_{\cdot \wedge \alpha W_z^\kappa}^{\overline{X^\epsilon}}$$

Convergence in Skorokhod topology does not automatically imply convergence of stopping times (see Appendix E.1 for a counterexample). However, for the high-probability event

$$C_{\kappa,z}^T = \left\{\omega \in \Omega : \forall\, t \in [0,T],\ B_t(\omega) \in \alpha W_z^\kappa\right\},$$

we have that $\exists \epsilon_0$ such that $\forall \epsilon > \epsilon_0$,

$$T_{\alpha W_z^\kappa}^{\overline{X^\epsilon}} \equiv T, \quad T_{\alpha W_z^\kappa}^B \equiv T,$$

and thus convergence holds on $C_{\kappa,z}^T$.

It remains to lower-bound $\mathbb{P}(C_{\kappa,z}^T)$. Observe that

$$\Omega \setminus C_{\kappa,z}^T \subset D_{\kappa,z}^T = \left\{\omega \in \Omega : \exists t \in [0,T],\ d_{M_z^\kappa}(B_t(\omega), z) \geq R_z^\kappa\right\}.$$

Hence

$$\mathbb{P}(C_{\kappa,z}^T) \ \geq \ 1 - \mathbb{P}(D_{\kappa,z}^T).$$

*Lemma* 1 (Exit-time bound). Suppose $L \geq 1$ satisfies

$$\sup_{x \in M_z^\kappa(\alpha)} \operatorname{Ric}_{M_z^\kappa}(x) \geq -L^2.$$

Let $T_{R_z^\kappa}$ be the first exit time of Riemannian Brownian motion from

$$B_{M_z^\kappa}(B_0, R_z^\kappa) = \{x \in M_z^\kappa(\alpha) : d_{M_z^\kappa}(B_0, x) < R_z^\kappa\}.$$

Then

$$\mathbb{P}(T_{R_z^\kappa} \leq T) \ \leq \ \exp\!\left(-\tfrac{(R_z^\kappa)^2}{8T}\left(1 - \tfrac{2Tk_z^\kappa L}{(R_z^\kappa)^2}\right)^2\right).$$

*Proof.* Let $r(x) = d_{M_z^\kappa}(B_0, x)$ and write $r_t := r(B_t)$. On $M_z^\kappa$ the function $r$ is smooth. The semimartingale decomposition of $r_t$ (see, e.g., Eq. 3.6.1 in (Hsu, 2002)) gives

$$r_t^2 \ \leq \ 2\int_0^t r_s\, d\beta_s \ + \ \int_0^t r_s\, \Delta r(B_s)\, ds \ + \ t,$$

where $\beta$ is a real Brownian motion adapted to $B$, and we have used that the local time term is nonnegative and can be dropped to obtain an inequality.

By the Laplacian comparison theorem $(\operatorname{Ricci}(\cdot) \geq -(k_z^\kappa - 1)L^2)$, for $r > 0$,

$$\Delta r \ \leq \ (k_z^\kappa - 1)\, L\, \coth(Lr) \ \leq \ (k_z^\kappa - 1)\!\left(L + \tfrac{1}{r}\right),$$

hence

$$r\,\Delta r \ \leq \ (k_z^\kappa - 1)\!\left(Lr + 1\right).$$

Up to the first exit time $T_{R_z^\kappa} := \inf\{t \geq 0 : r_t \geq R_z^\kappa\}$ we have $r_s \leq R_z^\kappa$, so

$$r_s\, \Delta r(B_s) \ \leq \ (k_z^\kappa - 1)\!\left(LR_z^\kappa + 1\right) \ \leq \ k_z^\kappa L + k_z^\kappa \ \leq \ 2\, k_z^\kappa L,$$

using $L \geq 1$. Therefore, for $t = T_{R_z^\kappa} \wedge T$,

$$r_t^2 \ \leq \ 2\int_0^t r_s\, d\beta_s \ + \ 2\, k_z^\kappa L\, t \ + \ t \ \leq \ 2\int_0^t r_s\, d\beta_s \ + \ 2\, k_z^\kappa L\, t \ + \ t. \tag{14}$$

On the event $\{T_{R_z^\kappa} \leq T\}$ we have $t = T_{R_z^\kappa}$ and $r_t \geq R_z^\kappa$, hence from (14)

$$(R_z^\kappa)^2 \ \leq \ 2\int_0^{T_{R_z^\kappa}} r_s\, d\beta_s \ + \ 2\, k_z^\kappa L T.$$

Rearranging,

$$\int_0^{T_{R_z^\kappa}} r_s \, d\beta_s \;\geq\; \frac{(R_z^\kappa)^2 - 2\,k_z^\kappa L\,T}{2}.$$

Set $M_t := \int_0^t r_s \, d\beta_s$, a continuous martingale with quadratic variation $\langle M \rangle_t = \int_0^t r_s^2 \, ds \leq (R_z^\kappa)^2 t$ up to time $T_{R_z^\kappa}$. By the Dambis–Dubins–Schwarz theorem there exists a standard Brownian motion $W$ such that $M_{T_{R_z^\kappa}} = W_\eta$ with $\eta = \langle M \rangle_{T_{R_z^\kappa}} \leq (R_z^\kappa)^2 T_{R_z^\kappa} \leq (R_z^\kappa)^2 T$ on $\{T_{R_z^\kappa} \leq T\}$. Thus,

$$\{T_{R_z^\kappa} \leq T\} \;\subset\; \left\{ W_\eta \;\geq\; \frac{(R_z^\kappa)^2 - 2\,k\kappa(z)L\,T}{2} \right\}.$$

Using the Gaussian tail bound together with $\eta \leq (R_z^\kappa)^2 T$ (and the reflection principle),

$$\mathbb{P}\big(T_{R_z^\kappa} \leq T\big) \;\leq\; \exp\left( -\frac{\big((R_z^\kappa)^2 - 2\,k_z^\kappa L\,T\big)^2}{8\,(R_z^\kappa)^2\,T} \right) \;=\; \exp\left( -\frac{(R_z^\kappa)^2}{8T}\left( 1 - \frac{2\,k_z^\kappa L\,T}{(R_z^\kappa)^2} \right)^2 \right),$$

which is the claimed bound. □

Applying the lemma, if $T < \frac{(R_z^\kappa)^2}{4k_z^\kappa L}$, then

$$\mathbb{P}(D_{\kappa,z}^T) = \mathbb{P}(T_{R_z^\kappa} \leq T) \;\leq\; \exp\left( -\frac{(R_z^\kappa)^2}{32T} \right),$$

which yields

$$\mathbb{P}(C_{\kappa,z}^T) \;\geq\; 1 - \exp\left( -\frac{(R_z^\kappa)^2}{32T} \right).$$

□

**Remark.**  The lemma shows that the probability of exiting the ball of radius $R_z^\kappa$ before time $T$ decays exponentially in $\frac{(R_z^\kappa)^2}{T}$, up to curvature- and rank-dependent constants. Intuitively, this means that with overwhelming probability the Riemannian Brownian motion (and hence our random walk in the $\epsilon \to 0$ limit) remains confined inside $B_{M_z^\kappa}(z, R_z^\kappa)$ for all $t \leq T$. This high-probability control is what allows us to restrict attention to the event $C_{\kappa,z}^T$ in the proof of Theorem 2.1.

### E.1. On stopping times and Skorohod convergence

An important subtlety in the proof of Theorem 2.1 is that convergence of processes in the Skorohod topology does not, in general, imply convergence of associated stopping times. Figure 7 illustrates this phenomenon with a simple deterministic example.

Let $X_t = \sin(t)$, and consider the approximating sequence of processes

$$X_t^n = \left( 1 - \frac{1}{n} \right) \sin(t).$$

Clearly $X^n \to X$ uniformly on compact time intervals, hence also in the Skorohod topology. However, the stopping time defined as

$$T = \inf\{t \geq 0 : X_t = 1\}$$

does not converge along this sequence. Indeed, $T = \pi/2$ for $X$, but for every finite $n$, the process $X^n$ never reaches 1 and therefore $T^n = \infty$. Thus, despite $X^n \to X$ in Skorohod topology, we have $T^n \not\to T$.

In the proof of Theorem 2.1 we avoid this issue by restricting to the high-probability set $C_{\kappa,z}^T$ where the Brownian path remains in the interior of the ball. On this event the stopping times agree with $T$, ensuring consistency with the limiting process.

## F. SORBES implementation details

We now provide practical details for the implementation of the SORBES algorithm.

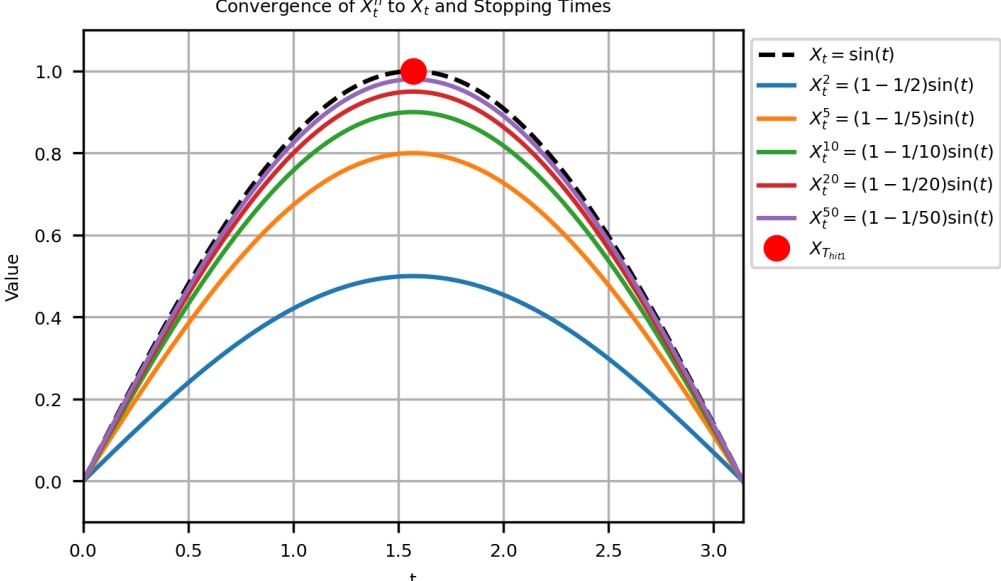

*Figure 7.* **Skorohod convergence does not imply convergence of stopping times.** The black dashed curve shows $X_t = \sin(t)$, which reaches 1 at $t = \pi/2$ (red dot). The colored curves show approximations $X_t^n = (1 - \frac{1}{n})\sin(t)$, converging uniformly to $X_t$. However, none of the $X_t^n$ ever reach 1, so their stopping times for hitting level 1 are infinite. This illustrates that Skorohod convergence of processes does not guarantee convergence of stopping times defined by hitting closed sets.

**Approximating Jacobian.**    The decoder Jacobian $J_{\hat{\mathrm{Dec}}}$ can be obtained exactly using one decoder forward pass and $LA$ backward passes, where $LA$ is a dimensionality of ambient space. To reduce computational cost, we approximate it via finite differences. Specifically, the $i$-th column of the Jacobian is approximated as

$$\frac{\partial \hat{\mathrm{Dec}}}{\partial z_i}(z) \approx \frac{\hat{\mathrm{Dec}}(z + \varepsilon e_i) - \hat{\mathrm{Dec}}(z)}{\varepsilon}, \tag{15}$$

where $e_i$ denotes the $i$-th standard basis vector in the latent space $\mathcal{Z}$ and $\varepsilon > 0$ is a small perturbation parameter. This approximation requires only $d+1$ decoder forward passes, where $d$ is a dimensionality of latent space, providing a substantial reduction in computational overhead.

**Sampling a unit tangent direction.**    We adopt the efficient implementation of Schwarz et al. (2023), which exploits a thin SVD of the decoder Jacobian to orthogonalize tangent directions.

**Approximating $\Gamma(z)[v, v]$.**    Computing Christoffel symbols directly requires evaluating first derivatives of the metric, which is computationally expensive and numerically unstable. Instead, we use an extrinsic approach: the covariant derivative of a curve can be obtained from its Euclidean acceleration in the ambient space, projected back onto the tangent space (Flaherty & do Carmo, 2013). Concretely, for a point $z \in \mathcal{Z}$, $z' \in W_z^\kappa$ and a probe radius $\rho > 0$, we approximate the extrinsic acceleration of a decoded curve along $v \in T_{z'} M_z^\kappa$ by a second-order central difference:

$$a_{\mathrm{ex}}^{z,\kappa}(v; z', \rho) \approx \frac{\hat{\mathrm{Dec}}(z' + \rho v) - 2\hat{\mathrm{Dec}}(z') + \hat{\mathrm{Dec}}(z' - \rho v)}{\rho^2}. \tag{16}$$

Projecting back to the latent tangent space using the Moore–Penrose pseudoinverse of the truncated Jacobian yields an efficient approximation of the Christoffel correction:

$$\Gamma(z')[v, v] \approx c(v; z', \rho) = J_{\hat{\mathrm{Dec}}}^\kappa(z')^+ a_{\mathrm{ex}}(v; z', \rho), \qquad c(v; z', \rho) \in \mathbb{R}^{k_z^\kappa}. \tag{17}$$

**Adaptive step size $\epsilon$.**    Since the computation of Christoffel symbols involves the (pseudo)inverse of the Jacobian, small values of $\kappa$ may amplify numerical noise. In this case, the update

$$z_i^\epsilon = z_{i-1}^\epsilon + \epsilon v - \epsilon^2 \Gamma[v, v] = z_{i-1}^\epsilon + \Delta(\epsilon),$$

may become unreasonably large in the ambient Euclidean metric on $U_z^\kappa$, making the algorithm unstable.

To control this, we adapt the step size $\epsilon$ so that the update norm never exceeds a predefined threshold $\Delta_{\max}$:

$$\epsilon_{\Delta_{\max}} = \min\left\{ \epsilon' > 0 : \|\Delta(\epsilon')\| \geq \Delta_{\max} \right\} \vee \epsilon,$$

where $a \vee b = \max\{a, b\}$. This guarantees that the step size is never greater than the nominal $\epsilon$, but shrinks adaptively whenever the second-order correction is large.

**SORBES (Stable/Efficient).**   We refer to the resulting algorithm with adaptive step size control and extrinsic approximation of Christoffel symbols (Section 2.3) as SORBES-SE. This variant is numerically stable in ill-conditioned regions of the decoder geometry while preserving the efficiency of the original scheme.

---

**Algorithm 4** SORBES-SE

---

**Require:** $z \in \mathcal{Z}$, $\kappa \geq 0$, step size $\epsilon$, diffusion time $T$, maximum number of steps $\texttt{STEP}_{\max}$, $\alpha = 0.99$, stability threshold
$\quad\quad \Delta_{\max} = 0.5$
1: $W_z^\kappa, G_{\text{Dec}} \leftarrow M_z^\kappa$
2: $z_0^\epsilon \leftarrow z,$
3: $\texttt{stopped} \leftarrow \texttt{False},$
4: $\sigma \leftarrow 0,$ $\hspace{8cm}$ ($\sigma$ tracks diffusion time)
5: $\texttt{step} = 0,$
6: **while** $\sigma < T$ and $\texttt{step} < \texttt{STEP}_{\max}$ **do**
7: $\quad$ Sample a unit tangent direction $\overline{v} \in S_z^\kappa = \{u \in T_z M_z^\kappa : \langle u, u \rangle_z^{\text{Dec}} = 1\}$
8: $\quad$ Set $v \leftarrow \sqrt{k_z^\kappa}\, \overline{v}$
9: $\quad$ **if** not $\texttt{stopped}$ **then**
10: $\quad\quad$ Compute trial update $\Delta(\epsilon) \leftarrow \epsilon v - \epsilon^2 \Gamma(z)[v, v]$
11: $\quad\quad$ **Adaptive adjustment:** If $\|\Delta(\epsilon)\| > \Delta_{\max}$, shrink step size:

$$\epsilon \leftarrow \min\{ \epsilon' > 0 : \|\Delta(\epsilon')\| \leq \Delta_{\max} \}$$

$\quad\quad$ and recompute $\Delta(\epsilon)$.
12: $\quad\quad$ Update latent coordinate:
$$z_i^\epsilon \leftarrow z_{i-1}^\epsilon + \Delta(\epsilon)$$
13: $\quad\quad$ $\sigma \leftarrow \sigma + \epsilon^2$ $\hspace{7cm}$ (update of diffusion time)
14: $\quad\quad$ **if** $z_i^\epsilon \notin W_z^\kappa(\alpha)$ **then**
15: $\quad\quad\quad$ $\texttt{stopped} \leftarrow \texttt{True}$
16: $\quad\quad$ **end if**
17: $\quad$ **else**
18: $\quad\quad$ $z_i^\epsilon \leftarrow z_{i-1}^\epsilon$ $\hspace{8cm}$ (absorbing state)
19: $\quad$ **end if**
20: $\quad$ $\texttt{step} \leftarrow \texttt{step} + 1$ $\hspace{8cm}$ (step update)
21: **end while**
$\quad$ **return** $(z_i^\epsilon)_{0 \leq i \leq \texttt{step}}$, $\sigma$

---

## G. MUTANG - Mutation Enumeration in Tangent Space

The detailed description of MUTANG algorithm is presented in Algorithm 5.

## H. Computational and Theoretical analysis of Local Enumeration

### H.1. Computational analysis of LOCALENUMERATION

Because SORBES-SE uses a finite-difference approximation, we can estimate the inference-time cost of Local Enumeration as a function of peptide length and latent dimensionality. Let $t_{\text{decoder}}$ denote the decoder forward-pass time, $k$ the latent

dimension, $n$ the maximum peptide length, and $a$ the alphabet size. The pessimistic computational cost of the Local Enumeration (LE) step can then be approximated as follows:

- Finite-difference evaluations for the Jacobian ($k$ evaluations) and the Christoffel symbols (2 evaluations): $(k+2)$, $t_{\text{decoder}}$.

- Singular value decomposition (SVD): $\mathcal{O}(kna)$.

- Projection of the extrinsic acceleration (second-order correction): $\mathcal{O}(kna)$.

- MUTANG computation: $\mathcal{O}(kn)$.

Overall, a Local Enumeration step scales linearly with the latent dimension $k$, peptide length $n$, and decoder evaluation time $t_{\text{decoder}}$.

### H.2. Theoretical analysis of LOCALENUMERATION

Although Theorem 2.1 proves an asymptotic convergence in the Skorohod topology, it does not guarantee convergence for a single SORBES-SE step. Rather, it provides principled support for the local steps used in LOCAL ENUMERATION. While each iteration applies one step, the full algorithm forms a sequence of such steps, effectively simulating a random walk over a union of manifolds.

Rigorous formulations of random walks in sub-Riemannian settings remain limited to structured cases(Lou, 2019; **?**; Herrmann et al., 2023), which do not capture the unstructured geometry of deep generative decoders. Thus, our implementation serves as a principled practical approximation, further strengthened by the generator convergence (Schwarz et al., 2023).

---

**Algorithm 5** MUTANG

---

**Require:** latent $z \in \mathcal{Z}$; $\kappa \geq 0$; token threshold $\theta_{\text{tok}}$.
1: Set $U_z^\kappa, k_z^\kappa$ as in Equation 8
2: $\mathtt{p} \leftarrow \mathtt{p}(z)$
3: $\mathcal{P} \leftarrow \emptyset$
4: **for** $j = 1$ to $k_z^\kappa$ **do**
5:    $\Delta \text{Dec}^{(j)} \leftarrow \mathtt{reshape}(U^\kappa(z)_{:,j}, (L, A))$
6:    **for** $\ell = 1$ to $L$ **do**
7:      **for** each $a \in \mathcal{A}$ **do**
8:        **if** $\left| \Delta \text{Dec}_{\ell,a}^{(j)} \right| \geq \theta_{\text{tok}}$ **then**
9:          $\mathcal{P} \leftarrow \mathcal{P} \cup \{(\ell, a)\}$
10:        **end if**
11:      **end for**
12:    **end for**
13: **end for**
14: **for** $\ell = 1$ to $L$ **do**
15:    $S_\ell \leftarrow \{a : (\ell, a) \in \mathcal{P}\} \cup \{\mathtt{p}_\ell\}$        (identity included)
16: **end for**
17: $\mathcal{C}(\mathtt{p}(z)) \leftarrow \prod_{\ell=1}^{L} S_\ell$
   **return** $\mathcal{C}(\mathtt{p}(z))$

---

## I. POGS

Below we present the details of PoGS training and evaluation metrics:

PoGS hyperparameters:

- PoGS without potential: $\lambda = 0$ and $\mu = 0.1$,

- Full PoGS: $\lambda = 0.01$ and $\mu = 0.1$.

- All: $\theta_{\text{pot}} = 5$.

PoGS metrics:

- chord ambient length:

$$\sum_{k=0}^{N-1} \|X_{k+1} - X_k\|_2$$

- chord latent length:

$$\sum_{k=0}^{N-1} \|z_{k+1} - z_k\|_2$$

For computation of seeds and wells, we excluded first and last 20% of a peptide path were excluded to avoid trivial rediscovery.

For each pair, the chord length $N$ was determined dynamically as

$$N = \lfloor \rho \cdot \|z_a - z_b\|_2 \rfloor,$$

where $\rho$ is the point density hyperparameter (set to $\rho = 90$ in our experiments). This construction guarantees that longer trajectories in the latent space are sampled more densely than shorter ones, preserving a uniform resolution across geodesics of varying length. The geodesic points $\{z_i\}_{i=1}^n$ were optimized using the Adam optimizer with learning rate $\eta = 10^{-3}$ and weight decay $10^{-5}$. We applied a `ReduceLROnPlateau` scheduler, which decreased the learning rate by a factor of $0.8$ whenever no improvement in the loss was observed for a number of iterations equal to the patience hyperparameter. The endpoints $z_a$ and $z_b$ were kept fixed throughout the optimization by zeroing their gradients at every step.

---

**Algorithm 6** PoGS

---

**Require:** seeds $z_a, z_b$, potential function $\Phi$, nb of segments $N$, weights $\lambda, \mu$, steps $T$,
1: Initialize $z_0 \leftarrow z_a$, $z_N \leftarrow z_b$, $z_{1:N-1}$ by linear interpolation in latent space
2: **for** $t = 1$ to $T$ **do**
3:     $X_k \leftarrow \log(\text{Dec})(z_k)$ for $k = 0..N$
4:     Compute energy $\mathcal{E}_\lambda(Z)$ as in (6)
5:     Take a gradient step on $z_{1:N-1}$ to minimize $\mathcal{E}_\lambda(Z)$
6: **end for**
    **return** $\{\text{p}(z_k)\}_{k=0}^N$

---

## J. APEX-potential for PoGS

The APEX predictor (Wan et al., 2024) estimates minimum inhibitory concentration (MIC) values against 11 bacterial strains, but it operates on concrete peptide *sequences*. In Potential-Minimizing Geodesic Search (PoGS), optimization proceeds over *latent-space chords*, i.e., intermediate points $z$ that decode to *position-factorized distributions* over peptides rather than single sequences:

$$\text{Dec}(z) \in \mathbb{R}^{L \times A},$$

where $L$ is the maximum peptide length and $A = 21$ is the amino-acid alphabet augmented with padding. To enable PoGS, we first *distill* the sequence-level APEX potential into a surrogate that accepts peptide *distributions*.

**Dataset construction.** Peptides from the HydrAMP training set (Szymczak et al., 2023) were encoded into latent codes $z$. In order to obtain multiple distributions from a single peptide, we then created four clones $z'$ of latent codes $z$. We applied a 2×2 perturbation scheme: two clones were injected with Gaussian noise $N(0, 0.05)$, and two were left unchanged. Finally, these four latent codes were decoded to $\text{Dec}(z')$, using a softmax scaling with temperature of 1.0 for one pair (noisy and non-noisy) and a temperature of 1.5 to the other pair, resulting in four distributions per peptide. This yielded 1,060,000 peptide distributions in total. For each $\text{Dec}(z')$, we enumerated the $N = 20$ most-probable sequences

$$\big(P_0(z'), \ldots, P_{N-1}(z')\big) \quad \text{with probabilities} \quad \big(p_0(z'), \ldots, p_{N-1}(z')\big),$$

applied APEX to each $P_i(z')$ to obtain MIC *vectors* $\text{MIC}_{P_i(z')} \in \mathbb{R}^{11}$, and defined the distribution's *expected* MIC vector via the probability-weighted average

$$\Phi_{\text{MIC}}^{\text{true}}\big(\text{Dec}(z')\big) = \sum_{i=0}^{N-1} \text{MIC}_{P_i(z')} \cdot \frac{p_i(z')}{\sum_{j=0}^{N-1} p_j(z')} \ \in \ \mathbb{R}^{11}.$$

**Training protocol and standardization.**  We split the dataset into 80% train, 10% validation, and 10% test in such a way that no two sets contain distributions originating from the same peptide. Let $\mu, \sigma \in \mathbb{R}^{11}$ be the per-strain mean and standard deviation computed *on the training set*. Targets were z-scored componentwise:

$$y^{\text{z}} = \frac{y - \mu}{\sigma}.$$

We trained an encoder-only transformer that *operates on distributions* $\text{Dec}(z) \in \mathbb{R}^{L \times A}$ and predicts z-scored MIC vectors in $\mathbb{R}^{11}$:

$$\Phi_{\text{MIC}}^{\text{model}} : \ \text{Dec}(z) \ \mapsto \ \mathbb{R}^{11}.$$

Architecture: three transformer encoder layers (four heads), embedding dimension 128, feed-forward dimension 256, dropout 0.05. Optimization used Adam (learning rate $10^{-4}$) for 15 epochs with mean-squared error (MSE) loss on z-scored targets:

$$\mathcal{L}_{\text{MSE}} = \frac{1}{11} \left\| \Phi_{\text{MIC}}^{\text{model}}(\text{Dec}(z)) - y^{\text{z}} \right\|_2^2.$$

**Final potential used by PoGS.**  PoGS operates on flattened *log-probabilities*. Let $X \in \mathbb{R}^{L \cdot A}$ be the flattened log-probability vector. We reconstruct a valid distribution using PyTorch-style operations:

$$P(X) = \text{softmax}\big( X.\,\text{reshape}(L, A), \ \texttt{dim} = 1 \big) \ \in \ [0, 1]^{L \times A},$$

where `dim=1` is the amino-acid dimension. The surrogate outputs a z-scored MIC vector

$$\widehat{m}^{\text{z}}(X) = \Phi_{\text{MIC}}^{\text{model}}\big(P(X)\big) \in \mathbb{R}^{11}.$$

Restricting to the three target *E. coli* strains (index set $\mathcal{I}_{E.\,coli}$), the scalar property potential used by PoGS is

$$\Phi(X) = \mathbf{1}^{\top} \big[\widehat{m}^{\text{z}}(X)\big]_{\mathcal{I}_{E.\,coli}}.$$

# K. PoGS ablation study

## K.1. Analysis of results for different choice of metric and prototypes set

To assess sensitivity of PoGS to the choice of the metric as well as to the choice of prototypes, we compared our implementation with two alternatives: using amino-acid probabilities instead of logits, and straight (Euclidean) interpolation, on a new set of 60 prototypes (Table 3). Logits yielded better performance in terms of potential, counts of seeds and wells, supporting our choice. Importantly, both metrics outperformed straight interpolation, and PoGS achieved even better results on these new prototypes than those reported in Table 1. These results show that PoGS improves results regardless of the metric used or the prototype set.

*Table 3.* PoGS with the original metric compared to PoGS with a metric on decoded probabilities, as well as to straight interpolation, for a different set of prototypes than in Table 1.

| Method | Latent Length | Ambient Length (orig.) | Ambient Length (probs) | Peptide Path Length | Potential | Seeds | Wells |
|---|---|---|---|---|---|---|---|
| Straight interpolation | $6.50 \pm 1.34$ | $3448.7 \pm 403.8$ | $4.76 \pm 4.94$ | $47.0 \pm 20.9$ | $-1732.8 \pm 272.7$ | $117 \pm 42$ | $41 \pm 14$ |
| PoGS (with probs metric) | $11.48 \pm 3.36$ | $17227.93 \pm 147.8$ | $1.67 \pm 1.16$ | $42.7 \pm 18.1$ | $-1729.9 \pm 271.4$ | $176 \pm 43$ | $67 \pm 14$ |
| PoGS (original) | $14.12 \pm 3.07$ | $5001.4 \pm 236.7$ | $9.97 \pm 4.96$ | $63.3 \pm 29.3$ | $-2329.0 \pm 420.0$ | $188 \pm 29$ | $80 \pm 20$ |

## K.2. Analysis of results for different choice of potential

To evaluate PoGS's sensitivity to the choice of potential and its applicability to multi-objective settings, we applied it to jointly maximize two physicochemical properties by assigning weight $alpha$ to the hydrophobicity and $1 - \alpha$ to charge (Table K.2). Across all values of $\alpha$, PoGS outperformed straight-line (Euclidean) interpolation. For this experiment, we selected a new set of 60 prototypes from the GRAMPA and DRAMP datasets, demonstrating that PoGS achieves superior performance regardless of both the potential used and the prototype set.

| Method | Latent Length | Ambient Length | Peptide Path Length | Max Hydrophobicity | Max Charge | Maximized Multiobjective Potential |
|---|---|---|---|---|---|---|
| PoGS w. $\alpha = 0.9$ | $12.07 \pm 2.57$ | $4573.57 \pm 19$ | $66.86 \pm 24.14$ | $8.40 \pm 0.72$ | $15.81 \pm 2.31$ | $24.21 \pm 2.42$ |
| PoGS w. $\alpha = 0.5$ | $10.04 \pm 2.05$ | $3624.69 \pm 185.09$ | $57.66 \pm 27.08$ | $7.64 \pm 0.63$ | $15.75 \pm 2.15$ | $23.39 \pm 2.24$ |
| PoGS w. $\alpha = 0.1$ | $10.61 \pm 2.13$ | $3651.19 \pm 190.39$ | $55.93 \pm 25.80$ | $7.91 \pm 0.64$ | $14.76 \pm 2.26$ | $22.67 \pm 2.35$ |
| Straight line (Euclidean) | $6.51 \pm 1.34$ | $3448.70 \pm 403.80$ | $47.00 \pm 20.90$ | $7.90 \pm 0.64$ | $13.76 \pm 2.22$ | $21.66 \pm 2.35$ |

*Table 4.* Comparison of PoGS multiobjective optimization across different $\alpha$ values and Euclidean straight-line baselines.

## L. LE-BO Hyperparameters

We enumerate all hyperparameter values of our optimization algorithm LE-BO and all its sub-algorithms.

- Algorithm 3 LE-BO - Local Enumeration Bayesian Optimization

    - Trust region distance $d_{\text{trust}} = 2$.
    - Number of ROBOT evaluations per iteration $k_{\text{ROBOT}} = 3$.
    - Diversity threshold $d_{\text{ROBOT}} = 2$.
    - Following the approach of (Eberhardt et al., 2024), as a surrogate model, we use a Gaussian Process GP with the Tanimoto similarity kernel (Szedmak & Bach, 2020), applied to the MAP4 fingerprints of peptides (Capecchi et al., 2020).
    - Aquisition function GP.acquistion was chosen to be Log Expected Improvement.

- Algorithm 2 LOCALENUMERATION

    - $\kappa_{SORBES} = 0.01$.
    - $\kappa_{\text{MUTANG}} = 10^{-6}$.
    - Number of trajectories $M = 10$.
    - Walk time budget $T_{\text{walk}} = 0.1$.
    - Nominal step size $\epsilon = 0.1$.
    - Mutation threshold $\theta_{\text{mut}} = 10^{-6}$.

- A probe radius $\rho > 0$ in the second-order central difference approximation of the extrinsic acceleration (Equation 16) $\rho = 0.05$.

- Step of the finite-difference approximation of the decoder Jacobian (Equation 15) $\varepsilon = 0.05$.

## M. LE-BO time and memory profiling

To quantify the computational gains of our method, we measured the average runtime of LE-BO over 10 iterations and 6 seeds, and compared it to SAASBO (Eriksson & Jankowiak, 2021) under the same conditions (Table M). LE-BO required $8\times$ less time per iteration and used $1.5\times$ less memory. Moreover, Local Enumeration accounted for only 35% of the total runtime, underscoring its efficiency. Average execution time of a single optimization run of LE-BO with 1400 iterations was 1h20m ($\pm$30m). All computations were performed for the HydrAMP model with latent dimension 64 and measured on a Mac Mini M4 Pro machine with 24GB of RAM.

To further contextualize these results, Table 6 compares the end-to-end runtime of LE-BO against a diverse set of peptide optimization baselines across their respective hardware configurations. Despite operating in a geometry-aware latent optimization framework, LE-BO remains computationally competitive with substantially more expensive global optimization methods such as AdaLead (Sinai et al., 2020), BO(Sinai et al., 2020), PEX (Ren et al., 2022), relaxed CMA-ES (Hansen, 2023), DynaPPO (Angermueller et al., 2020), LaMBO-2 (Gruver et al., 2023), and GFN-AL(Jain et al., 2022), many of which require between two and four hours per optimization run. While lightweight mutation-based baselines such as Random Mutation exhibit substantially lower runtimes, they do not provide the same structured exploration capabilities. Importantly, LE-BO achieves this favorable computational profile while simultaneously improving optimization quality and candidate diversity, demonstrating that geometry-aware local exploration can remain practical even in high-dimensional biological design spaces.

| Method | Local Enumeration Time / Iteration (s) | Total Iteration Time (s) | Memory (MB) |
|---|---|---|---|
| LE-BO | $1.03 \pm 0.31$ | $2.89 \pm 2.21$ | $1490 \pm 210$ |
| SAASBO ((Eriksson & Jankowiak, 2021)) | N/A | $23.31 \pm 9.94$ | $2248 \pm 12$ |

*Table 5.* Runtime and memory comparison between LE-BO and SAASBO ((Eriksson & Jankowiak, 2021)).

*Table 6.* Runtime comparison of optimization methods across different hardware configurations. Machine specifications: M1 — AMD Ryzen Threadripper 3990X with RTX 2080 11GB; M2 — Mac Mini M4 Pro 24GB; M3 — Intel Xeon 6248R (96C/192T); M4 — Intel Xeon Gold 6226R (64C/128T) with Tesla V100 32GB.

| Method | Time |
|---|---|
| LE-BO (M1) | $1:07h \pm 14min$ |
| LE-BO (M2) | $1:20h \pm 30min$ |
| Random Mutation (M1) | $37s \pm 1s$ |
| AdaLead (M3) | $2:51h \pm 6min$ |
| CBAS (M4) | $1:49h \pm 5min$ |
| DynaPPO (M4) | $3:13h \pm 15min$ |
| BO (M3) | $1:55h \pm 22min$ |
| PEX (M3) | $3:49h \pm 24min$ |
| relaxed CMA-ES (M3) | $3:40h \pm 11min$ |
| GFN AL (M3) | $2:46h \pm 2min$ |
| Lambo2 (M1) | $9min\ 35s \pm 21s$ |

## N. Geometrical Analysis of LE-BO

Across both HydrAMP and PepCVAE, the effective rank of the latent space is consistently much lower than the nominal latent dimension (Figure 4A,B). Furthermore, the effective rank exhibits a strong positive correlation with peptide length (Figure 4C,D; Appendix B), indicating increasing geometric complexity for longer peptides.

Figure 8 analyzes how this latent dimensionality affects local candidate enumeration. LE-BO with SORBES-SE and mutation enabled (Figure 8A,E) identifies substantially more candidates in local enumeration as the $\kappa$-stable dimension increases, corresponding to longer peptides. In contrast, Euclidean-based search exhibits the opposite trend, producing fewer candidates in higher-dimensional regions (Figure 8B,C,F,G). Since the proposed optimization procedure relies on dense local peptide neighborhoods, this behavior explains the weaker empirical performance of Euclidean approaches.

Quantitatively, LE-BO with SORBES-SE and mutation generates significantly more candidates on average ($16,730 \pm 14,786$; Figure 8A,E) than Euclidean search with mutation ($442 \pm 151$; Figure 8B,F). Even without mutation, SORBES-SE yields higher candidate counts ($17 \pm 5$; Figure 8D,H) than its Euclidean counterpart ($13 \pm 5$; Figure 8C,G). Finally, the $\kappa$-stable latent dimension is strongly negatively correlated with peptide charge (Spearman $r = -0.55$) and aromaticity ($r = -0.57$; Figure 9). This suggests that peptides associated with higher latent dimensionality tend to exhibit higher solubility but reduced membrane affinity and antimicrobial activity. By enumerating substantially more candidates in these high-dimensional regions, LE-BO increases the likelihood that such unfavorable properties can be corrected during optimization.

## O. Diversity Analysis

To further assess the effect of geometry-aware exploration, we evaluated the diversity of generated peptide candidates using the average normalized Levenshtein distance computed over sets of five generated sequences. As summarized in Table 7, LE-BO consistently produces more diverse candidate sets than its Euclidean counterpart, achieving a higher average diversity score across benchmarks (0.59 vs. 0.55). These results suggest that the Riemannian formulation encourages broader exploration of peptide space while preserving meaningful structural variations, whereas Euclidean exploration tends to generate more locally concentrated candidate distributions.

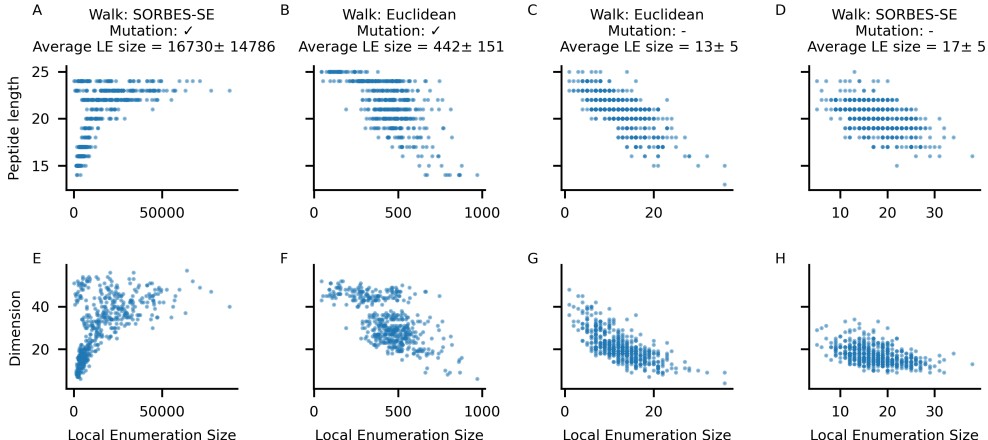

*Figure 8.* **Relationships between Local Enumeration (LE) result size and sequence properties under different ablations of LE-BO walk and mutation strategies**. Each panel (**A–H**) shows scatter plots comparing LE size with either peptide length (top row) or $\kappa$-stable dimension (bottom row) for four experimental LE-BO conditions: SORBES-SE walk with mutation (default LE-BO implementation) (**A, E**), Euclidean walk with mutation (**B, F**), Euclidean walk without mutation (**C, G**), and SORBES-SE walk without mutation (**D, H**). Titles report the mean ± standard deviation of the LE size for each condition. Together, these comparisons highlight how walk geometry and mutation choice influence the size and structure of local neighborhoods explored during Local Enumeration.

*Table 7.* Comparison of LE-BO and Euclidean LE-BO across peptide targets using the average normalized Levenshtein distance computed over sets of five generated candidates. LE-BO achieves higher diversity in four out of five benchmarks and improves the overall average diversity score from 0.55 to 0.59, indicating that geometry-aware exploration produces more diverse candidate sets than its Euclidean counterpart.

| | FL14 | KY14 | KK16 | mammuthusin-3 | hydrodamin-2 | Avg |
|---|---|---|---|---|---|---|
| Euclidean LE-BO | 0.56 | 0.54 | 0.54 | **0.63** | 0.50 | 0.55 |
| LE-BO | **0.59** | **0.57** | **0.58** | 0.60 | **0.60** | **0.59** |

# P. LE-BO Ablation study

## P.1. Analysis of results for different $\kappa_{SORBES}$ and $\kappa_{MUTANG}$

Additional ablation study show that alternative $\kappa$ values can even outperform those originally selected, demonstrating that LE-BO's performance can be further enhanced through targeted hyperparameter tuning *P*.1. An ablation with respect to $\alpha$ is unnecessary because, with only a single SORBES-SE step, $\alpha$ has no effect on the search process.

| $\kappa_{mutang}$ | $\kappa_{sorbes}$ | FL14 | KY14 | KF16 | KK16 | mammuthusin-3 | hydrodamin-2 | Avg. Diff. from LE-BO |
|---|---|---|---|---|---|---|---|---|
| $10^{-8}$ | $10^{-3}$ | $0.74 \pm 0.28$ | $0.83 \pm 0.44$ | $0.56 \pm 0.29$ | $0.58 \pm 0.30$ | **$0.44 \pm 0.29$** | **$0.38 \pm 0.13$** | $0.04 \pm 0.18$ |
| $10^{-4}$ | $10^{-3}$ | $0.53 \pm 0.26$ | $0.65 \pm 0.39$ | $0.66 \pm 0.11$ | **$0.36 \pm 0.09$** | $0.60 \pm 0.69$ | $0.45 \pm 0.14$ | **$-0.01 \pm 0.12$** |
| $10^{-8}$ | $10^{-1}$ | **$0.41 \pm 0.11$** | $0.65 \pm 0.44$ | **$0.38 \pm 0.14$** | $0.73 \pm 0.34$ | $0.58 \pm 0.21$ | $0.45 \pm 0.11$ | $-0.01 \pm 0.19$ |
| $10^{-4}$ | $10^{-1}$ | $0.83 \pm 0.20$ | $0.54 \pm 0.31$ | $0.72 \pm 0.29$ | $0.66 \pm 0.31$ | $0.60 \pm 0.45$ | $0.63 \pm 0.40$ | $0.12 \pm 0.07$ |
| $10^{-6}$ (orig.) | $10^{-2}$ (orig.) | $0.604 \pm 0.22$ | **$0.502 \pm 0.24$** | $0.600 \pm 0.29$ | $0.498 \pm 0.14$ | $0.498 \pm 0.38$ | $0.581 \pm 0.34$ | $0.0 \pm 0.0$ |

*Table 8.* LE-BO performance for different hyperparameter values. Best values of minimized MIC as predicted by APEX are bolded. The last column reports the average difference between the alternative hyperparameter setting and the original used in the manuscript.

## P.2. Analysis of results of LE-BO without ROBOT ((Maus et al., 2023))

Ablation analysis of the model without ROBOT (Maus et al., 2023) confirms that ROBOT improves LE-BO for 4 out of 6 seeds (with average difference in log MIC of -0.03).

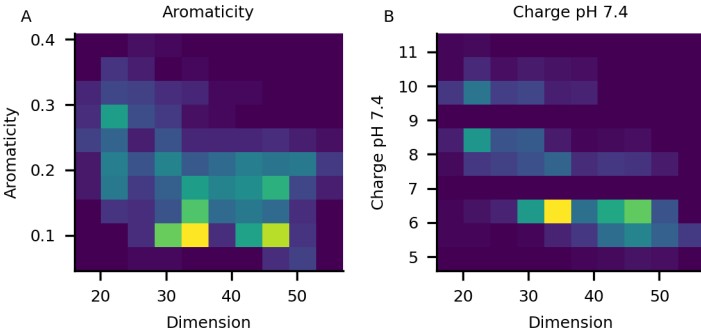

*Figure 9.* **Relationship between peptide physicochemical properties and latent dimension**. (**A**) Aromaticity versus embedding dimension, showing that peptides with lower aromatic residue content tend to occupy higher-dimensional regions of the embedding space. (**B**) Net charge at pH 7.4 versus embedding dimension, highlighting a similar trend in which peptides with lower cationic charge are more common at higher dimensions.

| Method | FL14 | KY14 | KF16 | KK16 | mammuthusin-3 | hydrodamin-2 | Avg. diff. from LE-BO |
|---|---|---|---|---|---|---|---|
| LE-BO w/o ROBOT | $0.75 \pm 0.41$ | $0.52 \pm 0.17$ | **$0.56 \pm 0.11$** | $0.63 \pm 0.28$ | $0.57 \pm 0.61$ | **$0.42 \pm 0.20$** | $0.03 \pm 0.12$ |
| LE-BO | **$0.604 \pm 0.22$** | **$0.502 \pm 0.24$** | $0.600 \pm 0.29$ | **$0.498 \pm 0.14$** | **$0.498 \pm 0.38$** | $0.581 \pm 0.34$ | **$0.000 \pm 0.000$** |

*Table 9.* Comparison of LE-BO with and without ROBOT across six peptides and mean deviation from LE-BO baseline.

## P.3. Analysis of results for other seeds

To show consistent gains compared to ablations across different seeds, we additionally performed an LE-BO ablation study on three further seeds (LL13, RC16, and KI21) from the PoGS optimization (Table P.3). The results confirm that clear advantage of enabling mutations and of using SORBES-SE as the random walk method can be observed regardless of the chosen seeds. Taken together, all three LE-BO components - SORBES-SE, enabling mutation (MUTANG), and ROBOT - contribute meaningful improvements. However, enabling mutation with MUTANG provides the most substantial benefit: across all ablations, the LE-BO variants incorporating MUTANG consistently outperform their counterparts without it.

| | Walk | Mutation | LL13 | RC16 | KI21 | Difference from LE-BO |
|---|---|---|---|---|---|---|
| | Euclidean | $\times$ | $0.787 \pm 0.408$ | $1.141 \pm 0.298$ | $1.170 \pm 0.283$ | $1.033 \pm 0.174$ |
| | SORBES-SE | $\times$ | $1.009 \pm 0.408$ | $1.128 \pm 0.298$ | $1.120 \pm 0.283$ | $0.709 \pm 0.204$ |
| | – | $\checkmark$ | $1.343 \pm 0.356$ | $1.225 \pm 0.275$ | $0.803 \pm 0.238$ | $0.747 \pm 0.125$ |
| | Euclidean | $\checkmark$ | $0.630 \pm 0.190$ | $0.751 \pm 0.291$ | $0.505 \pm 0.199$ | $0.253 \pm 0.091$ |
| LE-BO | SORBES-SE | $\checkmark$ | **$0.482 \pm 0.226$** | **$0.458 \pm 0.243$** | **$0.190 \pm 0.191$** | **$0.000 \pm 0.000$** |

*Table 10.* Confirmation of improved performance of LE-BO compared to ablations for different seeds than in Table 2

## P.4. Analysis of results for different oracles

To evaluate robustness across different oracles and to verify that additional peptide properties benefit from geometry-aware exploration, we successfully applied LE-BO and show that geometry-aware exploration improves prototype peptides for three tasks:

1. minimize MIC using DEEP-AMP (Pandi et al., 2023) regressor other than APEX as oracle,

2. minimize toxicity using ToxiPrep (Guan et al., 2025) classification probabilities as oracle,

3. maximize hydrophobicity computed in the Eisenberg scale (Eisenberg, David and Weiss, Robert M. and Terwilliger, Thomas C. and Wilcox, William, 1982) as oracle.

|  | KY14 | KF16 | KK16 | FL14 | mammuthusin-3 | hydrodamin-2 |
|---|---|---|---|---|---|---|
| $\log_2$(MIC) (seed) | 2.00 | 5.06 | 1.85 | 1.82 | 4.02 | 5.09 |
| Euclidean LE-BO | -0.66 ± 0.50 | 0.04 ± 0.19 | -0.56 ± 0.65 | -0.55 ± 0.46 | -0.48 ± 0.72 | -0.15 ± 0.56 |
| LE-BO | **-0.80 ± 1.06** | **-0.47 ± 0.38** | **-1.20 ± 1.23** | **-2.35 ± 0.41** | **-0.77 ± 0.75** | **-0.93 ± 0.89** |

*Table 11.* Minimization of $\log_2$(MIC) values using LE-BO with DEEP-AMP (Pandi et al., 2023) as oracle.

|  | KY14 | KF16 | KK16 | FL14 | mammuthusin-3 | hydrodamin-2 |
|---|---|---|---|---|---|---|
| Toxicity (seed) | 0.8789 | 0.9283 | 0.9044 | 0.9932 | 0.5193 | 0.0522 |
| Euclidean LE-BO | 0.014 ± 0.002 | 0.016 ± 0.003 | 0.016 ± 0.005 | 0.015 ± 0.001 | 0.014 ± 0.002 | 0.016 ± 0.002 |
| LE-BO | **0.0115 ± 0.0007** | **0.0117 ± 0.0012** | **0.0120 ± 0.0022** | **0.0122 ± 0.0015** | **0.0126 ± 0.0013** | **0.0135 ± 0.0012** |

*Table 12.* Minimization of toxicity values using LE-BO with toxicity probabilities returned by ToxiPep (Guan et al., 2025) as oracle.

|  | KY14 | KF16 | KK16 | FL14 | mammuthusin-3 | hydrodamin-2 |
|---|---|---|---|---|---|---|
| Hydrophobicity (seed) | $-0.34$ | $-0.60$ | 0.03 | 0.12 | 0.17 | $-0.33$ |
| Euclidean LE-BO | 1.114 ± 0.091 | 1.208 ± 0.048 | 1.162 ± 0.112 | 1.193 ± 0.029 | 1.323 ± 0.031 | **0.963 ± 0.084** |
| LE-BO | **1.255 ± 0.013** | **1.270 ± 0.048** | **1.242 ± 0.067** | **1.265 ± 0.026** | **1.325 ± 0.022** | 0.953 ± 0.095 |

*Table 13.* Maximization of hydrophobicity using LE-BO with hydrophobicity values computed in Eisenberg scale (Eisenberg, David and Weiss, Robert M. and Terwilliger, Thomas C. and Wilcox, William, 1982) oracle.

## Q. Wet-lab validation

### Q.1. Peptide selection

For PoGS, we selected the top 4 sequences with the highest predictive oracle scores from the global search (Section 3.1).

For LE-BO, we ran 10 independent optimizations per seed and extracted the top 10 peptides from each run, yielding 400 candidates. After applying synthesizability filters from our wet-lab partners, 62 were removed, leaving 338 candidates. These were then reviewed by experts, who selected final analogs based on viability, synthesis success, and structural novelty.

### Q.2. Peptide synthesis and characterization

Peptides were synthesized on an automated peptide synthesizer (Symphony X, Gyros Protein Technologies) by standard Fmoc-based solid-phase peptide synthesis (SPPS) on Fmoc-protected amino acid–Wang resins (100–200 mesh). The following preloaded resins were employed with their respective loading capacities (100 $\mu$mol scale): Fmoc-Asn(Trt)-Wang Resin (0.510 mmol g$^{-1}$), Fmoc-His(Trt)-Wang Resin (0.480 mmol g$^{-1}$), Fmoc-Leu-Wang Resin (0.538 mmol g$^{-1}$), Fmoc-Lys(Boc)-Wang Resin (0.564 mmol g$^{-1}$), Fmoc-Phe-Wang Resin (0.643 mmol g$^{-1}$), Fmoc-Thr(tBu)-Wang Resin (0.697 mmol g$^{-1}$), Fmoc-Trp(Boc)-Wang Resin (0.460 mmol g$^{-1}$), Fmoc-Tyr(tBu)-Wang Resin (0.520 mmol g$^{-1}$). In addition to preloaded resins, standard Fmoc-protected amino acids were employed for chain elongation, including: Fmoc-Ala-OH, Fmoc-Cys(Trt)-OH, Fmoc-Glu(OtBu)-OH, Fmoc-Phe-OH, Fmoc-Gly-OH, Fmoc-His(Trt)-OH, Fmoc-Ile-OH, Fmoc-Lys(Boc)-OH, Fmoc-Leu-OH, Fmoc-Met-OH, Fmoc-Asn(Trt)-OH, Fmoc-Arg(Pbf)-OH, Fmoc-Ser(tBu)-OH, Fmoc-Thr(tBu)-OH, Fmoc-Val-OH, Fmoc-Trp(Boc)-OH, and Fmoc-Tyr(tBu)-OH. N,N-Dimethylformamide (DMF) was used as the primary solvent throughout synthesis. Stock solutions included: 500 mmol L$^{-1}$ Fmoc-protected amino acids in DMF, a coupling mixture of HBTU (450 mmol L$^{-1}$) and N-methylmorpholine (NMM, 900 mmol L$^{-1}$) in DMF, and 20% (v/v) piperidine in DMF for Fmoc deprotection. After synthesis, peptides were deprotected and cleaved from the resin using a cleavage cocktail of trifluoroacetic acid (TFA)/triisopropylsilane (TIS)/dithiothreitol (DTT)/water (92.8% v/v, 1.1% v/v, 0.9% w/v, 4.8% w/w) for 2.5 hours with stirring at room temperature. The resin was removed by vacuum filtration, and the peptide-containing solution was collected. Crude peptides were precipitated with cold diethyl ether and incubated for 20 min at $-20\,^{\circ}$C, pelleted by centrifugation, and washed once more with cold diethyl ether. The resulting pellets were dissolved in 0.1% (v/v) aqueous formic acid and incubated overnight at $-20\,^{\circ}$C, followed by lyophilization to obtain dried peptides. For characterization, peptides were dried, reconstituted in 0.1% formic acid, and quantified spectrophotometrically. Peptide separations were performed on a Waters XBridge C$_{18}$ column (4.6 × 50 mm, 3.5 $\mu$m, 120 Å) at room temperature using a conventional high-performance liquid chromatography (HPLC) system. Mobile phases were water with 0.1% formic acid (solvent A) and acetonitrile with 0.1% formic acid (solvent B). A linear gradient of 1–95% B over 7 min was applied at 1.5 mL min$^{-1}$. UV detection was monitored at 220 nm. Eluates were analyzed on Waters SQ Detector 2 with electrospray

*Table 14.* Bacterial strains used for experimental validation of antimicrobial peptide libraries. Strains marked with MDR are multidrug-resistant clinical isolates.

| ID | Bacterial Strain |
| --- | --- |
| AB1 | *A. baumannii* ATCC 19606 |
| AB2$_{MDR}$ | *A. baumannii* ATCC BAA-1605 |
| EC1 | *E. cloacae* ATCC 13047 |
| EC2 | *E. coli* ATCC 11775 |
| EC3 | *E. coli* AIC221 |
| EC4$_{MDR}$ | *E. coli* AIC222 |
| EC5$_{MDR}$ | *E. coli* ATCC BAA-3170 |
| KP1 | *K. pneumoniae* ATCC 13883 |
| KP2$_{MDR}$ | *K. pneumoniae* ATCC BAA-2342 |
| PA1 | *P. aeruginosa* PAO1 |
| PA2 | *P. aeruginosa* PA14 |
| PA3$_{MDR}$ | *P. aeruginosa* ATCC BAA-3197 |
| SE1 | *S. enterica* ATCC 9150 |
| SE2 | *S. enterica Typhimurium* ATCC 700720 |
| BS1 | *B. subtilis* ATCC 23857 |
| SA1 | *S. aureus* ATCC 12600 |
| SA2$_{MDR}$ | *S. aureus* ATCC BAA-1556 |
| EFS1$_{MDR}$ | *E. faecalis* ATCC 700802 |
| EFU1$_{MDR}$ | *E. faecium* ATCC 700221 |

ionization in positive mode. Full scan spectra were collected over m/z 100–2,000. Selected Ion Recording (SIR) was used for targeted peptides. Source conditions were capillary voltage 3.0 kV, cone voltage 25–40 V, source temperature 120 °C, and desolvation temperature 350 °C. Mass spectra were processed with MassLynx software. Observed peptide masses were compared with theoretical values, and quantitative analysis was based on integrated SIR peak areas.

## Q.3. Bacterial Strains and Growth Conditions

The bacterial panel utilized in this study consisted of the following pathogenic strains (see Table 14: *Acinetobacter baumannii* ATCC 19606; *A. baumannii* ATCC BAA-1605 (resistant to ceftazidime, gentamicin, ticarcillin, piperacillin, aztreonam, cefepime, ciprofloxacin, imipenem, and meropenem); *Escherichia coli* ATCC 11775; *E. coli* AIC221 [MG1655 phnE_2::FRT, polymyxin-sensitive control]; *E. coli* AIC222 [MG1655 pmrA53 phnE_2::FRT, polymyxin-resistant]; *E. coli* ATCC BAA-3170 (resistant to colistin and polymyxin B); *Enterobacter cloacae* ATCC 13047; *Klebsiella pneumoniae* ATCC 13883; *K. pneumoniae* ATCC BAA-2342 (resistant to ertapenem and imipenem); *Pseudomonas aeruginosa* PAO1; *P. aeruginosa* PA14; *P. aeruginosa* ATCC BAA-3197 (resistant to fluoroquinolones, $\beta$-lactams, and carbapenems); *Salmonella enterica* ATCC 9150; *S. enterica* subsp. *enterica* Typhimurium ATCC 700720; *Bacillus subtilis* ATCC 23857; *Staphylococcus aureus* ATCC 12600; *S. aureus* ATCC BAA-1556 (methicillin-resistant); *Enterococcus faecalis* ATCC 700802 (vancomycin-resistant); and *Enterococcus faecium* ATCC 700221 (vancomycin-resistant). *P. aeruginosa* strains were propagated on Pseudomonas Isolation Agar, whereas all other species were maintained on Luria-Bertani (LB) agar and broth. For each assay, cultures were initiated from single colonies, incubated overnight at 37 °C, and subsequently diluted 1:100 into fresh medium to obtain cells in mid-logarithmic phase.

## Q.4. Minimal Inhibitory Concentration (MIC) determination

MIC values were established using the standard broth microdilution method in untreated 96-well plates. Test peptides were dissolved in sterile water and prepared as twofold serial dilutions ranging from 1 to 64 $\mu$mol L$^{-1}$. Each dilution was combined at a 1:1 ratio with LB broth containing $4 \times 10^6$ CFU mL$^{-1}$ of the target bacterial strain. Plates were incubated at 37 °C for 24 h, and the MIC was defined as the lowest peptide concentration that completely inhibited visible bacterial growth. All experiments were conducted independently in triplicate.

## Q.5. Detailed wet-lab validation results

Q.5.1. COMPOSITION OF THE EXPERIMENTALLY VALIDATED PEPTIDE SET

We experimentally validated 3 types of peptide sets:

- **Prototypes (P1, P2)**: Known AMPs that serve as anchors for geodesic interpolation. These are not novel peptides and therefore not counted among the 29 experimentally evaluated peptides.
- **Seeds (S)**: Novel peptides discovered via PoGS through geodesic interpolation between prototype pairs. We identified 4 such seeds for experimental validation.
- **Analogs** ($A_1 - A_n$): Novel peptides derived from each seed through LE-BO optimization. We generated a total of 25 analogs across the 4 seed families.

The 25 analogs generated for the experimental validation were distributed as follows:

- Seed-1 $\rightarrow$ 7 analogs (LE-BO-1-1 -LE-BO-1-7)
- Seed-2 $\rightarrow$ 6 analogs (LE-BO-2-1 -LE-BO-2-6)
- Seed-3 $\rightarrow$ 6 analogs (LE-BO-3-1 -LE-BO-3-6)
- Seed-4 $\rightarrow$ 6 analogs (LE-BO-4-1 -LE-BO-4-6)

Q.5.2. CROSS-METHOD SUCCESS RATE COMPARISON ON SHARED BACTERIAL PANELS

Prior wet-lab validated AMP generation methods were evaluated against heterogeneous bacterial panels, so a fair comparison is difficult to achieve and the success rates can be inflated due to differences in strain susceptibility. To address this, we recomputed success rates on strains shared across methods. On the 11-strain panel shared with AMP-Diffusion (the only prior method with substantial overlap with our broader panel), PepCompass reaches a $100\%$ success rate at MIC $\leq 8\,\mu$M against at least one strain, whereas AMP-Diffusion plateaus near $76\%$ even at MIC $\leq 128\,\mu$M. Restricting further to *E. coli* strains shared with HydrAMP, AMP-Diffusion, CLaSS, and Joker yields a comparable picture: PepCompass attains $100\%$ success at MIC $\leq 32\,\mu$M, while the strongest baseline reaches $92\%$ only at MIC $\leq 64\,\mu$M, and CLaSS remains below $15\%$ across all thresholds. Full numerical curves are reported in Figure 10.

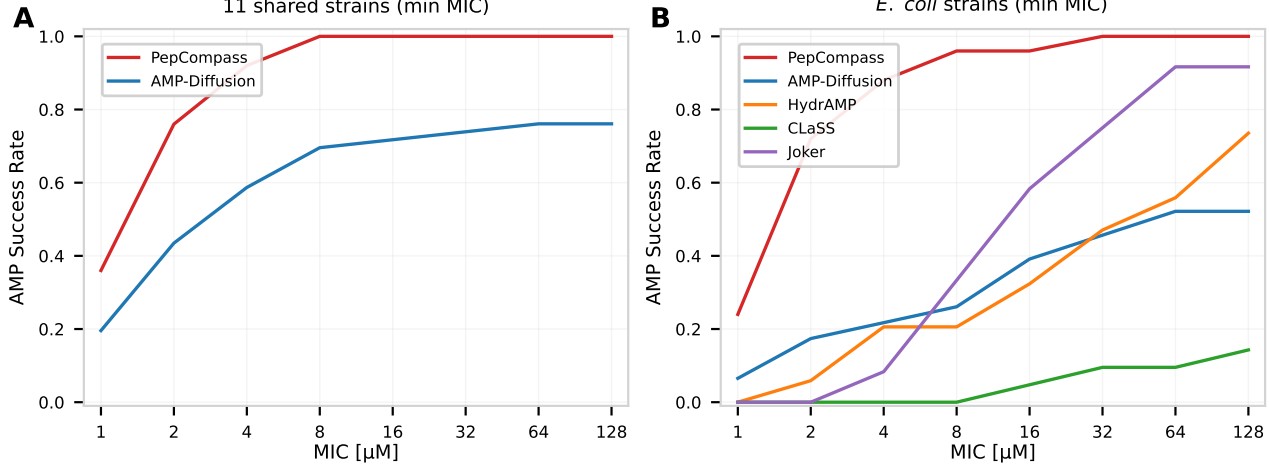

*Figure 10.* **AMP success rates restricted to bacterial strains shared across methods.** (A) Success rate against the 11 strains shared between PepCompass and AMP-Diffusion. (B) Success rate against *E. coli* strains shared with HydrAMP, AMP-Diffusion, CLaSS, and Joker. Success rate is defined as the fraction of generated peptides with MIC below the specified threshold against at least one strain in the restricted panel.

*Table 15.* Minimum inhibitory concentration (MIC, in $\mu$mol L$^{-1}$) values and peptide sequences for PepCompass-generated peptides tested against bacterial pathogen panel. '-' indicates MIC $>64$ $\mu$mol L$^{-1}$. Strain IDs correspond to Table 14.

| ID | Sequence | AB1 | AB2 | EC1 | EC2 | EC3 | EC4 | EC5 | KP1 | KP2 | PA1 | PA2 | PA3 | SE1 | SE2 | BS1 | SA1 | SA2 | EFS1 | EFU1 |
|---|---|---|---|---|---|---|---|---|---|---|---|---|---|---|---|---|---|---|---|---|
| Prototype-1-a | ILRWKKRKLVWKR | 64 | - | - | 16 | 64 | 32 | - | - | - | 8 | 32 | - | - | - | - | 32 | 32 | - | - |
| Prototype-1-b | FLILRWSRFARVLL | 8 | - | - | 64 | 8 | 16 | - | - | - | - | - | - | - | - | - | 16 | 8 | 32 | 4 |
| Seed-1 | FLYKWWIRIGRLKL | 1 | 1 | - | 32 | 4 | 8 | 16 | - | - | - | - | - | 32 | 8 | 16 | - | - | 64 | 64 |
| LE-BO-1-1 | RYAKINLRTAWRKLKWLIKKVMKKW | 4 | 4 | 64 | 8 | 4 | 4 | 4 | 8 | 32 | 8 | 4 | 4 | 2 | 2 | - | - | 32 | - | - |
| LE-BO-1-2 | RYAKINLRTAWRKLKWLIKKVMKWW | 8 | 8 | 64 | 8 | 4 | 16 | 8 | 8 | 8 | 8 | 8 | 8 | 4 | 4 | 64 | 16 | 32 | - | - |
| LE-BO-1-3 | RKANLKSRYAWLKLRKLIKALIAWK | 2 | 4 | - | 8 | 4 | 4 | 4 | 4 | 8 | 8 | 4 | 4 | 2 | 2 | - | 32 | 64 | - | - |
| LE-BO-1-4 | RKANLKSRYAWLKLRKLIKALVAWK | 1 | 16 | 32 | 4 | 2 | 4 | 4 | 4 | 8 | 8 | 4 | 4 | 2 | 2 | - | 32 | - | - | - |
| LE-BO-1-5 | RKANLKSRYAWLKLRKLIKAVILWK | 4 | 8 | - | 16 | 8 | 8 | 4 | 16 | 8 | 8 | 4 | 8 | 4 | 2 | - | - | - | - | - |
| LE-BO-1-6 | RKANLKTRYAWLKLRKLIKAVVNWK | 1 | 2 | - | 16 | 4 | 8 | 4 | 4 | 8 | 8 | 4 | 2 | 2 | 2 | - | 64 | - | - | - |
| LE-BO-1-7 | RKANLKIRYAWLKLRNLIKAAINWK | 1 | 1 | - | 4 | 4 | 4 | 2 | 2 | 2 | 8 | 8 | 16 | 2 | 4 | - | - | 16 | - | - |
| Prototype-2-a | KFWARGRKPWKLAIQILK | 4 | - | - | 8 | 4 | 2 | - | - | - | - | 16 | - | - | - | - | - | - | - | 4 |
| Prototype-2-b | ILRWKKRWKVWLR | 8 | - | - | 2 | 2 | 1 | - | - | - | 2 | 8 | - | - | - | - | 16 | 16 | - | - |
| Seed-2 | KFRNRHRWKFKLIFRN | 4 | 8 | - | 16 | 16 | 32 | 8 | - | - | 8 | 8 | 16 | 8 | 4 | - | - | - | - | 16 |
| LE-BO-2-1 | NRRKYLRYWLKKLLRKILKAAINAW | 8 | 8 | - | 16 | 4 | 16 | 16 | 4 | 8 | 8 | 16 | 8 | 4 | 4 | - | - | - | - | - |
| LE-BO-2-2 | KARIKLYYRWKLKLKWLLKAMIKAW | 8 | 8 | - | 16 | 4 | 16 | 8 | 8 | 4 | 8 | 16 | 64 | 8 | 32 | 32 | - | - | - | - |
| LE-BO-2-3 | KARIKLRYRWKLKLKWLLKMAAMAW | 1 | 1 | 64 | 2 | 1 | 2 | 2 | 1 | 1 | 2 | 1 | 4 | 1 | 1 | - | - | - | - | - |
| LE-BO-2-4 | KARIKLRYRWKLKLKWLLKAMMAAW | 2 | 1 | - | 2 | 2 | 2 | 4 | 4 | 8 | 4 | 2 | 4 | 4 | 4 | - | - | - | - | - |
| LE-BO-2-5 | KARIKLRYRWRLKLKWLLKMAWAAW | 32 | 64 | - | 64 | 32 | 64 | 16 | 16 | 64 | 16 | 64 | - | - | - | - | - | - | - | - |
| LE-BO-2-6 | KARIKLRYRWRLKLKWLLKAMFAW | 4 | 16 | - | 16 | 8 | 16 | 16 | 16 | - | - | 16 | 16 | 16 | 64 | - | - | - | - | - |
| Prototype-3-a | ILRWKFRKWVWLR | 4 | - | - | 2 | 8 | 2 | - | - | - | 8 | 8 | - | - | - | - | 16 | 32 | - | - |
| Prototype-3-b | WRHKSLWIRKYLKNLALLA | 0.78 | - | - | 3.12 | 0.78 | 1.56 | - | 25 | - | 6.25 | - | - | - | - | - | 50 | 25 | - | 3.12 |
| Seed-3 | KKYWLIRKWIRLWFLT | 16 | 32 | - | - | 16 | 32 | 8 | 64 | 64 | - | 32 | 64 | 16 | 32 | 16 | 64 | - | - | - |
| LE-BO-3-1 | KKARNLRKWAYLKYRLKLKILAINW | 32 | 64 | - | - | 8 | - | 64 | 64 | - | 64 | - | 64 | - | - | - | - | - | - | - |
| LE-BO-3-2 | KKARNLRWKAYLKYRLKLKILAWNK | 8 | 8 | - | 64 | 64 | - | - | - | - | - | - | 32 | 64 | - | - | - | - | - | - |
| LE-BO-3-3 | KKRRKLTLKLKLKKLLRLL | 2 | 1 | 64 | 2 | 2 | 4 | 4 | 64 | 8 | 4 | 1 | 4 | 4 | 8 | - | - | - | - | - |
| LE-BO-3-4 | KKARNLRKWAYLKYRLKLKILAANW | 32 | 32 | - | - | 4 | - | 16 | 64 | - | - | 16 | 32 | 64 | - | - | - | - | - | - |
| LE-BO-3-5 | KLRISLKARWRLWKMYVLKWKAAIW | - | - | - | 32 | 2 | 16 | 64 | 16 | 16 | 64 | - | - | 64 | - | - | - | - | - | - |
| LE-BO-3-6 | KKRRILRKWTRLWKKLLELMAAWFH | 8 | - | - | 32 | 4 | 4 | 8 | - | 8 | 32 | 8 | 8 | 8 | 16 | - | - | - | - | - |
| Prototype-4-a | LIRWKVRWLAFRRL | 4 | - | - | 4 | 8 | 8 | - | 16 | - | 16 | 16 | - | - | - | - | - | 64 | 32 | 1 |
| Prototype-4-b | KYCWRWFKLLFKKL | 4 | - | - | 4 | 2 | 2 | - | 8 | - | - | 32 | - | - | - | - | 16 | 16 | - | 4 |
| Seed-4 | KYCRRFRWLTFRWL | 64 | 32 | - | 16 | 8 | 8 | 16 | 16 | 32 | 64 | 64 | 64 | 16 | 16 | 64 | - | - | - | - |
| LE-BO-4-1 | KRARNYYRWKLWKKLKILLKAAMAW | 2 | 2 | - | 8 | 4 | 4 | 8 | 2 | 4 | 8 | 8 | 8 | 8 | 8 | - | 32 | - | - | 32 |
| LE-BO-4-2 | KRIRKLRILRTWKWWKLEMAAAFH | 16 | 4 | - | 64 | 8 | 16 | 32 | 32 | - | 64 | 16 | 32 | 4 | 32 | 64 | - | - | - | 32 |
| LE-BO-4-3 | KRLRKLRILRTWKWWKLEMAAAFH | 16 | 16 | - | 32 | 16 | 16 | 16 | 16 | - | 64 | 32 | 16 | 2 | 8 | - | - | - | - | 64 |
| LE-BO-4-4 | KRIRKLRILRTWKWWKLEMAMAFH | 16 | 8 | - | 64 | 16 | 4 | 8 | 16 | - | 64 | 16 | 64 | 4 | 8 | 64 | - | - | - | - |
| LE-BO-4-5 | KRLRKLRILRTWKWWKLEMAAAFHY | 32 | 16 | - | - | 8 | 16 | 16 | 64 | - | - | 16 | 16 | 4 | 16 | - | - | - | - | - |
| LE-BO-4-6 | KRLRKLRILRTWKWWKLEMAAAFHF | 4 | 4 | - | 16 | 8 | 4 | 16 | 16 | - | - | - | 16 | 4 | - | - | - | - | - | 64 |

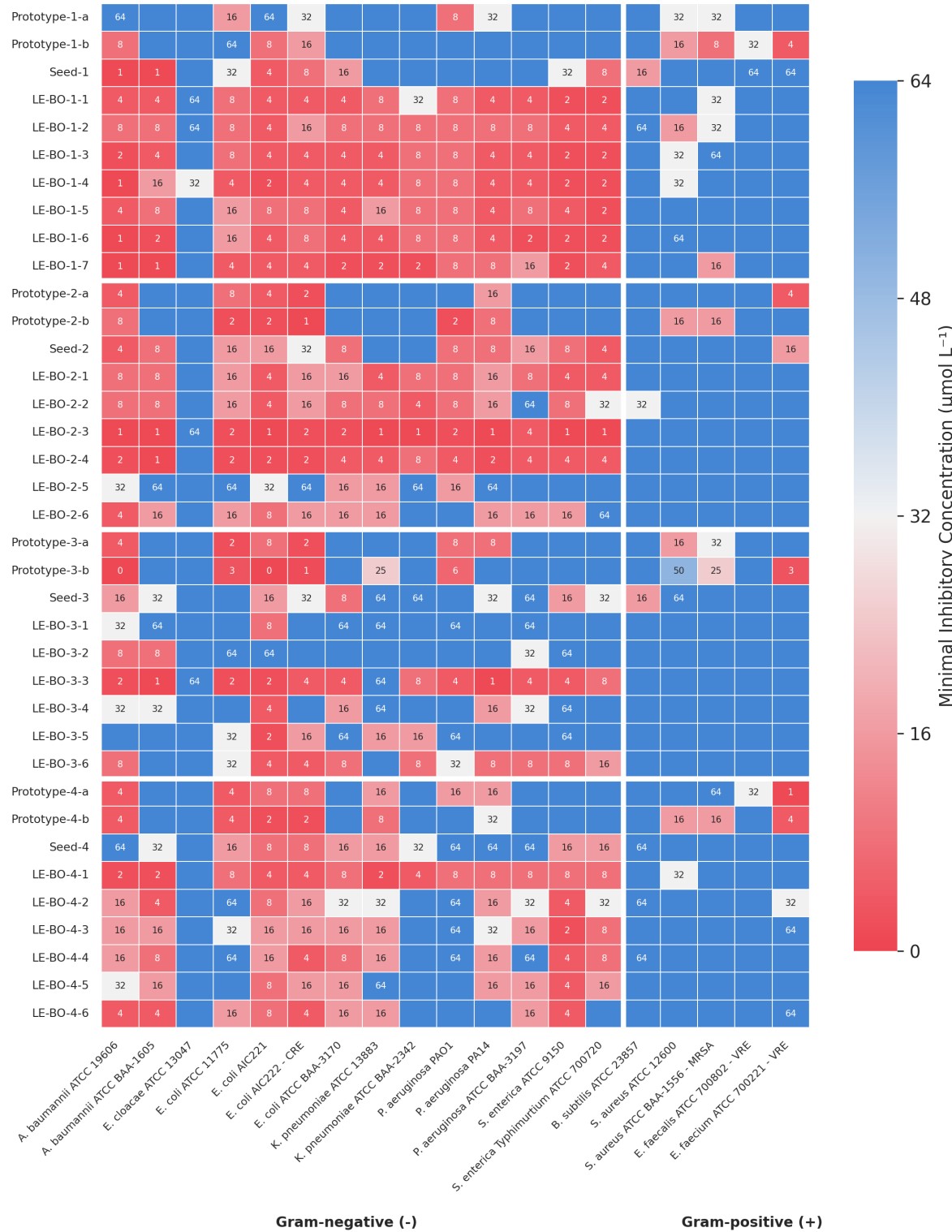

*Figure 11.* **Minimum inhibitory concentration profiles of antimicrobial peptide libraries against Gram-negative and Gram-positive bacterial pathogens**. MIC values (in $\mu$mol L$^{-1}$) for 37 peptide sequences evaluated against 19 bacterial strains. Peptides are stratified by seed family (seed-1 through seed-4), comprising parental prototype sequences and corresponding analogs generated via LE-BO. The bacterial panel encompasses 14 Gram-negative strains including carbapenem-resistant *Enterobacteriaceae* (CRE: *E. coli* AIC222 and ATCC BAA-3170), extended-spectrum $\beta$-lactamase-producing *K. pneumoniae* (ATCC BAA-2342), and fluoroquinolone-resistant *P. aeruginosa* (ATCC BAA-3197), alongside 5 Gram-positive strains including methicillin-resistant *S. aureus* (MRSA: ATCC BAA-1556) and vancomycin-resistant *Enterococcus* species (VRE: *E. faecalis* ATCC 700802, *E. faecium* ATCC 700221).

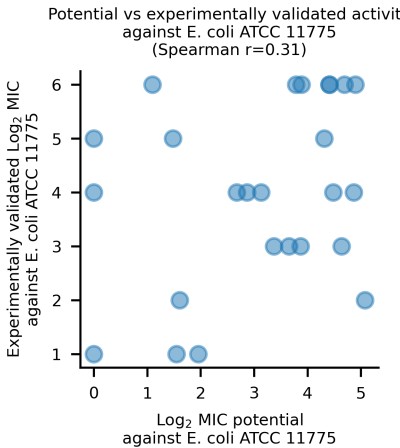

*Figure 12.* **Correlation between APEX potential and experimental antimicrobial activity for 29 validated peptides.** LE-BO–predicted $Log_2$ MIC potential shows a meaningful positive association with experimentally measured $Log_2$ MIC against *E. coli* ATCC 11775 (Spearman $r = 0.31$).

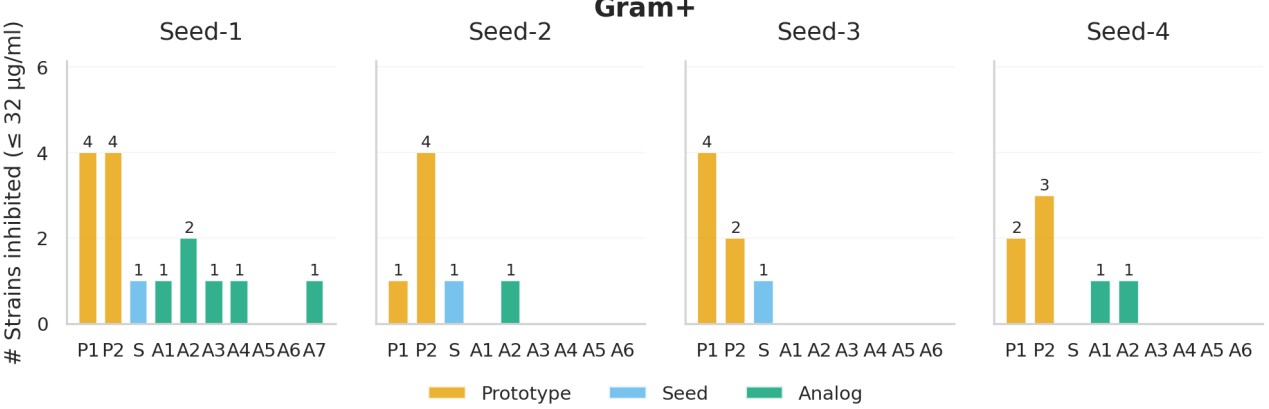

*Figure 13.* **Antimicrobial activity against Gram-positive bacterial strains by seed family.** Bar chart shows the number of Gram-positive strains (out of 5 total) against which each peptide achieved MIC $\leq 32$ $\mu$g/ml, organized by seed family (Seed-1 through Seed-4). Within each family, results are shown for prototypes (P1, P2; orange), seeds (S; blue), and analogs (A1-A7; green). Numbers above bars indicate the count of active strains for each peptide.

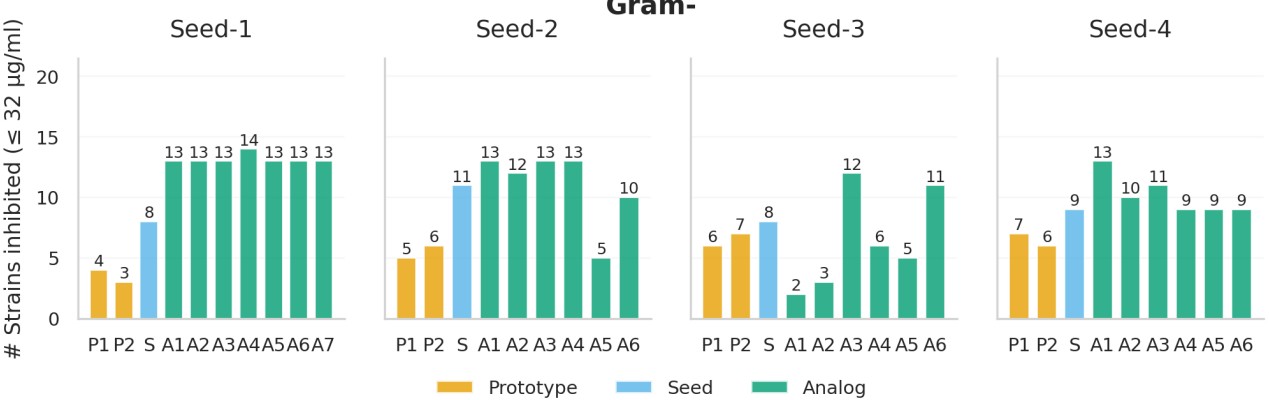

*Figure 14.* **Antimicrobial activity against Gram-negative bacterial strains by seed family.** Bar chart shows the number of Gram-negative strains (out of 14 total) against which each peptide achieved MIC $\leq$ 32 $\mu$g/ml, organized by seed family (Seed-1 through Seed-4). Within each family, results are shown for prototypes (P1, P2; orange), seeds (S; blue), and analogs (A1-A7; green). Numbers above bars indicate the count of active strains for each peptide.

