# OpenReview forum: "PepCompass: Navigating Peptide Embedding Spaces Using Riemannian Geometry"
_ICML.cc/2026/Conference — ICML 2026 regular_

### Official Review · Reviewer_CVAQ · 2026-03-10

**Soundness:** 3
**Presentation:** 3
**Significance:** 3
**Originality:** 3
**Overall Recommendation:** 4
**Confidence:** 3

**Summary:**

This work proposes PepCompass, a geometry-aware exploration framework on latent representations of peptide sequences, with an application to antimicrobial peptide (AMP) discovery. The paper propose modeling the latent space of pre-trained peptide generative models as a union of κ-stable Riemannian manifolds. Based on this formulation, PepCompass provides peptide exploration and optimization at both global and local levels. At the global level, Potential-minimizing Geodesic Search (PoGS) interpolates between prototype peptide pairs along paths biased toward high predicted antimicrobial activity. At the local level, a second-order Riemannian Brownian motion approximation (SORBES) and tangent-space mutation enumeration (MUTANG) are combined into an iterative Local Enumeration procedure, which is further integrated with Bayesian optimization (LE-BO). Wet-lab validation on 29 peptides demonstrates a 100% success rate at the 32 µg/ml MIC threshold, including activity against multidrug-resistant strains.

**Compliance With Llm Reviewing Policy:**

Affirmed.

**Final Justification:**

The authors' rebuttal adequately addressed my main concerns, including the two major weaknesses (W1, W2) raised in the initial review. I maintain my score of 4 (weak accept), which was already leaning toward acceptance. The paper makes a solid contribution to geometry-aware peptide optimization with strong wet-lab validation that others are likely to build on.

**Key Questions For Authors:**

**Q1**. The Introduction presents PepCompass as a framework built upon the union of κ-stable Riemannian manifolds, yet PoGS operates in the ambient Euclidean space without relying on this formulation, and MUTANG appears to depend primarily on Jacobian SVD truncation rather than the Riemannian manifold structure. Among the three main components, only SORBES seems to require the κ-stable formulation, yet the ablation suggests its contribution to final performance is marginal. Could the authors clarify how the κ-stable union of manifolds formulation concretely underpins PoGS and MUTANG, or whether the framework's practical contribution is largely independent of this formulation?

**Q2**. Theorem 2.1 establishes convergence in the ε → 0 limit, but only a single SORBES-SE step is used in practice. Could you clarify whether the theoretical guarantees of Theorem 2.1 apply in any meaningful way to this single-step regime, or whether the theorem should be understood as an independent theoretical contribution separate from the practical algorithm?

**Limitations:**

Yes

**Strengths And Weaknesses:**

## Strengths

**S1. Importance of the problem and generality of the approach.**
The paper addresses the practically relevant problem of exploring and optimizing AMP sequences within the latent space of generative models. The proposed framework is not limited to peptides; the κ-stable manifold formulation and Local Enumeration strategy could in principle extend to other domains where data lies on a union of manifolds and efficient exploration of design candidates is required. The 100% wet-lab validation rate across 29 peptides, including activity against multidrug-resistant strains, further demonstrates the practical strength of the approach.

**S2. Well-grounded theoretical contribution.**
The paper presents a rigorous mathematical framework, including the formal construction and maximality proof of κ-stable Riemannian manifolds, a convergence theorem extending prior work (Schwarz et al., 2023) to non-compact manifolds, and a formal analysis of the mutational geometry induced by the ambient metric.

**S3. Comprehensive experimental evaluation.**
The experimental evaluation includes comparisons against 16+ baselines and extensive ablation studies covering walk type, mutation enumeration, κ values, alternative seeds, and alternative oracles.

**S4. Empirical validation of varying-dimension latent structure.**
The paper clearly demonstrates that the effective latent dimension is consistently below the nominal dimension and correlates with peptide length, providing concrete empirical support for the union-of-manifolds motivation.

**S5. Methodological novelty.**
The paper introduces several novel components: SORBES as a second-order convergent approximation to Riemannian Brownian motion, MUTANG as an interpretable bridge between continuous tangent-space geometry and discrete mutations, and PoGS as a bi-prototype interpolation scheme with property-aware potential.

## Weaknesses

**W1. The practical effect of κ-stable manifold formulation appears marginal.**
The Euclidean walk + MUTANG ablation achieves performance within one standard deviation of the full LE-BO on most seeds, suggesting that MUTANG is the major driver of improvement rather than the Riemannian walk. Moreover, MUTANG uses very small κ and θ_mut values that admit nearly all Jacobian entries, making it unclear how much the κ-stable structure contributes beyond simple SVD truncation. PoGS also operates in the ambient Euclidean space without using the κ-stable formulation. Among the three main components, only SORBES requires the κ-stable manifold structure, yet the ablation suggests that SORBES contributes only marginally to the final performance. I would encourage the authors to clarify what concrete benefits the κ-stable union of manifolds formulation provides beyond enabling SORBES.

**W2. Theorem 2.1 assumptions do not match the actual use case.**
The theorem establishes convergence in the limit ε → 0, requiring unbounded steps T/ε^2, but in practice only a single SORBES-SE step is used. The gap between the theoretical guarantees and the practical implementation should be discussed more explicitly.

**W3. Missing explanation of wet-lab selection criteria; potential selection bias.**
The criteria for selecting 29 peptides from the large pools of candidates generated by PoGS and LE-BO are not provided. Appendix P.1 ("Experimental setup") is empty. Without this information, it is difficult to assess whether selection bias may have contributed to the reported success rate.

**W4. Presentation issues.**
(a) The Introduction should more clearly separate the pullback metric (the solution to flat-metric distortion) from the manifold hypothesis (its precondition for well-definedness). (b) The claim that the manifold hypothesis implies full-rank Jacobian implicitly assumes d equals the intrinsic data dimension; this should be stated explicitly. (c) A high-level overview of the full pipeline (PoGS → LE-BO) before describing individual components would improve readability.

**W5. Lack of justification for MUTANG's candidate construction.**
MUTANG constructs candidates as the Cartesian product of per-position admissible sets, implicitly assuming positional independence. This choice is not justified, and no ablation compares it against simpler alternatives such as single-site mutations. The first-order tangent-space analysis may not reliably predict the combined effect of multiple simultaneous mutations.

---

> ### Author Rebuttal · Authors · 2026-03-30
>
> We thank the reviewer for the insightful feedback. Below, we address each of the concerns in detail.
>
> > Q1. The Introduction presents PepCompass as a framework built upon the union of κ-stable Riemannian manifolds, yet PoGS operates in the ambient Euclidean space without relying on this formulation, and MUTANG appears to depend primarily on Jacobian SVD truncation rather than the Riemannian manifold structure. Among the three main components, only SORBES seems to require the κ-stable formulation, yet the ablation suggests its contribution to final performance is marginal. Could the authors clarify how the κ-stable union of manifolds formulation concretely underpins PoGS and MUTANG, or whether the framework's practical contribution is largely independent of this formulation?
> W1 [..] Moreover, MUTANG uses very small κ [..] values that admit nearly all Jacobian entries, making it unclear how much the κ-stable structure contributes beyond simple SVD truncation.
>
> We thank the reviewer for the opportunity to clarify the role of the κ-stable formulation. While Euclidean walk + MUTANG performs competitively, Appendix O shows a key difference: combining SORBES with MUTANG yields far more valid candidates during local enumeration (16,730 vs 442; Fig. 8A–B), driven by improved exploration of high-dimensional latent regions.
>
> Regarding SVD truncation, even for very small κ (e.g., 10⁻⁸), the local κ-stable dimension remains substantially lower than the latent dimensionality (avg. 12.17 vs 64; Appendix Fig. 4). Thus, the κ-stable formulation is not incidental—it ensures numerical stability in SORBES by removing degenerate directions and governs the sensitivity–accuracy trade-off in MUTANG.
>
> Regarding PoGS we refrained from using a κ-stable formulation as its usage would require solving a computationally heavy differential geodesic equations.
>
> > W2, Q2. Theorem 2.1 proves convergence as ε → 0 with many steps, whereas the method uses only a single SORBES-SE step in practice. Could the authors clarify whether these guarantees meaningfully apply to the single-step regime [..]?
>
> We agree that Theorem's 2.1 asymptotic convergence in the Skorohod topology does not guarantee convergence for a single SORBES-SE step. Rather, it provides principled support for the local steps used in Local Enumeration. While each iteration applies one step, the full algorithm forms a sequence of such steps, effectively simulating a random walk over a union of manifolds.
>
> Rigorous formulations of random walks in sub-Riemannian settings remain limited to structured cases (e.g., [[1](https://arxiv.org/pdf/1604.07870), [2](https://www.sciencedirect.com/science/article/pii/S0304414918303065), [3](https://arxiv.org/abs/2311.17289)]), which do not capture the unstructured geometry of deep generative decoders. Thus, our implementation serves as a principled practical approximation, further strengthened by the generator convergence in $L^{\infty}$ [[4](https://arxiv.org/pdf/2202.00959)]. We will clarify this theory–practice gap in the Appendix.
>
> > W3.Missing explanation of wet-lab selection criteria; potential selection bias.
>
> For PoGS, we selected the top 4 sequences with the highest predictive oracle scores from the global search (Section 3.1).
>
> For LE-BO, we ran 10 independent optimizations per seed and extracted the top 10 peptides from each run, yielding 400 candidates. After applying synthesizability filters from our wet-lab partners, 62 were removed, leaving 338 candidates. These were then reviewed by experts, who selected final analogs based on viability, synthesis success, and structural novelty.
>
> We will describe this pipeline in details, in the revised Appendix P.1 for full transparency.
>
> > W4. Presentation issues.
>
> We thank the reviewer for these constructive suggestions, which will improve the manuscript’s readability and clarity. We will incorporate them in the camera-ready version. Specifically, we will revise the Introduction to clarify the theoretical preconditions of the pullback metric and our assumptions on intrinsic dimensionality, and expand Figure 1’s caption to provide a clear high-level overview of the full pipeline.
>
> > W5. MUTANG factorization issues
>
> We agree that constructing candidates via a Cartesian product relies on a first-order approximation and assumes positional independence, which we will explicitly acknowledge as a limitation. Extending MUTANG to capture multi-site dependencies is an important direction for future work. We also note that MUTANG naturally includes single-site mutations via the identity token in each position, and our experiments already compare against a purely single-site baseline (“Random Mutation”).
>
> ___
> We will incorporate these clarifications in the final version of the manuscript to further strengthen the presentation and positioning of our contributions. We hope these revisions address the reviewer’s concerns and kindly ask them to consider increasing their score in light of these improvements.

---

> > ### Author Rebuttal · Reviewer_CVAQ · 2026-04-02
> >
> > Thank you for the authors’ response. I will keep my score, which is already leaning toward acceptance.

---

### Official Review · Reviewer_uoqH · 2026-03-13

**Soundness:** 3
**Presentation:** 3
**Significance:** 3
**Originality:** 3
**Overall Recommendation:** 4
**Confidence:** 3

**Summary:**

This paper introduces PepCompass, a geometry-aware framework for peptide (AMP) discovery that models the latent space of a model as a union of Riemannian manifolds. They find property-optimizing paths between peptides as geodesics in these manifolds. They conducted wet-lab validation of designed peptides..

**Compliance With Llm Reviewing Policy:**

Affirmed.

**Final Justification:**

The authors have thoroughly answered my questions. I lean towards acceptance.

**Key Questions For Authors:**

1. How sensitive are the results to the choice of κ-stability threshold? Could you provide a sensitivity analysis or a principled method for selecting κ?
2. Have you considered replacing the Euclidean ambient-space metric with biologically informed metrics (e.g., BLOSUM-derived)? What would be the computational overhead?
3. What is the computational cost of PepCompass (wall-clock time per peptide designed) relative to simpler baselines like directed evolution with ESM scoring?
4. Were any toxicity or hemolysis assays performed on the 29 synthesized peptides? If not, do you have plans for such validation?
5. How does the manifold structure (number of patches, dimensionality) vary across the 6 prototype peptide families? Is there evidence that the framework adapts well to structurally diverse peptide types?

**Limitations:**

The Euclidean metric assumption in the ambient space is the most strange choice for me. The lack of toxicity/selectivity data limits the practical impact claims. Scalability beyond short AMPs is unaddressed. The framework requires a pretrained VAE with a differentiable decoder, which constrains the choice of base model.

**Strengths And Weaknesses:**

Strengths:
- The formalization of the latent space as a union of κ-stable Riemannian manifolds seems mathematically rigorous (though I'm not an expert), and the convergence guarantee for SORBES (Theorem 2.1) seems to be an original contribution. The use of the decoder Jacobian to define the pullback metric and SVD-based κ-stability criteria is well-motivated and elegant.
- Solid experimental validation.
- Thorough baselines. Table 2 compares LE-BO against 15+ methods across 6 prototype peptides, providing a broad competitive landscape. The ablation study clarifies the contribution of each component.
- Extensive appendix. Proofs (Appendix E), construction details (Appendix C), and mutational metric interpretation (Appendix D) provide substantial supplementary material for reproducibility and verification. Though I confess I did not have time to check all the proofs thoroughly in detail.

Weaknesses:
- The authors acknowledge that they use the standard Euclidean metric in the ambient (sequence) space when computing the pullback. For discrete objects like peptide sequences, this could potentially be improved. The paper does not explore how sensitive results are to this choice, e.g., using a BLOSUM substitution matrix could be more biologically principled.
- The peptides studied are relatively short AMPs. The computational cost of computing the full decoder Jacobian and performing SVD grows with sequence length. The paper does not discuss scalability to longer sequences.
- The choice of κ (the condition number threshold for defining stable manifold patches) appears to require fine-tuning, but the paper does not provide clear guidance on how κ should be selected for new encoder architectures or different peptide families.
- Missing toxicity/hemolysis data. For AMP design, selectivity (antimicrobial activity vs. host cell toxicity) is a critical concern. The paper reports MIC values but does not present hemolysis or cytotoxicity assays, leaving the therapeutic relevance of the designed peptides somewhat incomplete. This could also be left for a future study.

---

> ### Author Rebuttal · Authors · 2026-03-30
>
> We thank the reviewer for the insightful feedback. Below, we address each of the concerns in detail.
>
> > W1 The authors acknowledge that they use the standard Euclidean metric in the ambient (..). The paper does not explore how sensitive results are to this choice, e.g., using a BLOSUM substitution matrix could be more biologically principled.
>
> Thank you for this insightful comment. We agree that replacing the Euclidean metric with a biologically informed versions is a promising direction. As discussed in Appendix D, our current choice serves as a mathematically sound, non-informative, and conservative baseline. Regarding metrics like BLOSUM, we outline the mathematical integration of such biologically meaningful metrics in Appendix D.3. However as standard BLOSUM matrices are derived from general proteins, they may not capture the specific biological intricacies of AMP optimization, we are actively developing task-specific metrics to be released in future work. Importantly, the computational overhead is minimal: it requires a one-time SVD of the kernel matrix offline and a single additional fully connected layer applied to the decoder output.
>
> > W2. (..) The paper does not discuss scalability to longer sequences.
>
> PepCompass assumes a constant-size tensor of factorized probabilities (with padding for shorter sequences), so the baseline cost per model is fixed, regardless of peptide length. Appendix H shows that the method scales linearly with a maximal peptide length, latent dimensionality, and decoder evaluation time.
>
> > W3, Q1 The choice of κ (..) appears to require fine-tuning, but the paper does not provide clear guidance on how κ should be selected for new encoder architectures or different peptide families.
>
> The stability of our method with respect to the choices of κ is analyzed in Appendix N.1 and Table 6. The ablation study demonstrates that the model's performance is highly stable across various κ values. The only exception occurs when the κ are set to excessively high values ($κ_{MUT}=10^{−4}$ and $κ_{SORBES}=10^{−1}$), which performs significantly worse. Thus, for the fine-tuning of kappa for a given architectural setup and task, we would recommend a typical cross validation setup running over a grid of kappa values used in Appendix N.1 and Table 6, excluding those excessive values.
>
> > W4, Q4. Missing toxicity/hemolysis data. For AMP design, selectivity (antimicrobial activity vs. host cell toxicity) is a critical concern.
>
> We fully agree that hemolysis and cytotoxicity analyses are crucial for evaluating the clinical potential of these peptides. However, given the scope and constraints of a conference submission, our *in vitro* validation focused strictly on antimicrobial activity (MIC). We are currently planning a follow-up study covering the host toxicity panels to better understand the therapeutic viability of the generated sequences.
>
> > Q3. What is the computational cost of PepCompass (wall-clock time per peptide designed) relative to simpler baselines like directed evolution with ESM scoring?
>
> We performed additional analysis for directed evolution with ESM scoring (referred to as ESM Random Mutation) on the KY14 seed, that performed worse then previously benchmarked directed evolution (Random Mutation): log MIC 1.88±0.01 vs 1.23±0.26. Below we present the wall-clock time of the selected methods, averaged across 5 runs (with 1400 steps and surrogate model evaluations/retrainings) for KY14. Due to limited rebuttal time, these benchmarks were performed on multiple machines (M1-M4), with their specification outline below the table. These specifications should be accounted for when comparing runtimes between the methods.
>
> |Method|Time
> |-|-|
> | LE-BO (M1)|1:07h±14min
> | LE-BO (M2)|1:20h±30min
> | ESM Random Mutation (M1)|70s±7s
> | Random Mutation (M1)|37s±1s
> | AdaLead (M3)|2:51h±6min
> | CBAS (M4)|1:49h±5min
> | DynaPPO (M4)|3:13h±15min
> | BO (M3)| 1:55h±22min
> | PEX (M3)|3:49h±24min
> | relaxed CMA-ES (M3)|3:40h±11min
> | GFN AL (M3)|2:46h±2min
> | Lambo2 (M1)|9min 35s±21s
>
> M1: AMD Ryzen Threadripper 3990X RTX2080 11GB
>
> M2: Mac Mini M4Pro 24GB
>
> M3: Intel Xeon 6248r 96C/192T
>
> M4: Intel Xeon Gold 6226r 64C/128T, Tesla V100 32GB
>
> > Q5. How does the manifold structure (..) vary across the 6 prototype peptide families? Is there evidence that the framework adapts well to structurally diverse peptide types?
>
> We provide the dimensionality distribution for different seed families in a new [figure](https://ibb.co/W4jWkDLF) attached. In Appendix O, we present how our approach adapts to the stability, charge and aromaticity of the modelled peptide,  to improve its optimization properties.
>
> ---
> We will incorporate these clarifications and additional analyses in the final version of the manuscript to further strengthen the presentation and positioning of our contributions. We hope these revisions address the reviewer’s concerns and kindly ask them to consider increasing their score in light of these improvements.

---

> > ### Author Rebuttal · Reviewer_uoqH · 2026-03-31
> >
> > The authors have thoroughly answered my questions. I retain my score but increase my confidence.

---

### Official Review · Reviewer_vcF5 · 2026-03-13

**Soundness:** 2
**Presentation:** 1
**Significance:** 3
**Originality:** 3
**Overall Recommendation:** 4
**Confidence:** 2

**Summary:**

This paper proposes PepCompass, a geometry-aware framework for navigating peptide embedding spaces for antimicrobial peptide design. The method models the latent space as a union of κ-stable Riemannian manifolds and introduces several components for exploration and optimization in this space. Experimental results are reported on peptide design tasks, and the authors further conduct wet-lab validation to test the biological activity of generated peptides.

**Compliance With Llm Reviewing Policy:**

Affirmed.

**Final Justification:**

The authors have fully addressed my comments, and the revision is now much clearer.

**Key Questions For Authors:**

1. Could the authors clarify the problem formulation and the specific task the proposed framework is designed to solve?
2. Could the authors provide more intuition and background explanations for the main components of the method?

**Limitations:**

Yes.

**Strengths And Weaknesses:**

# Strengths
- The work proposes a geometry-based framework for latent space exploration in peptide design.
- The study includes wet-lab experimental validation, which strengthens the empirical impact of the work.

# Weaknesses
- Problem formulation is not sufficiently clear or complete. The paper does not clearly define the precise problem setting or the motivation behind the proposed formulation.
- Background and intuition are insufficient. Many steps of the method are introduced with mathematical definitions, but without sufficient explanation of the intuition.
- Some descriptions are unclear. For example, the statement “A common approach …” (around line 66) does not clearly specify a common approach to what problem.
- Assumptions (e.g., the full-rank assumption) are introduced without adequate explanation.

---

> ### Author Rebuttal · Authors · 2026-03-30
>
> We thank the reviewer for the insightful feedback. Below we clarify the problem formulation and provide intuition for the method, and will revise the manuscript accordingly.
>
> ## Problem Formulation
>
> Our goal is to address the problem of ***efficient exploration and optimization of peptide space using latent generative models***.
>
> More concretely, given:
>
> - a trained generative model for peptide sequences with latent space $Z$ and decoder $Dec$ that maps points from latent space $Z$ to ambient space $X$ (in our case - space of continuous approximations of peptide sequences),
> - a target biological property (in our case - antimicrobial activity),
>
> we aim to ***navigate the latent space in a way that produces valid peptides while efficiently optimizing the target property.***
>
> The key challenge is that:
>
> - common approaches treat $Z$ as **Euclidean** space and use a naive standard Euclidean metric for the analysis and optimization, including methods for dimensionality reduction like PCA, UMAP or T-SNE, for clustering like K-Means, for Bayesian optimization: Gaussian / Mattern kernels, Gaussian / Normal sampling used in exploration,
> - but as the points from $Z$ pass through the decoder, this induces a **non-Euclidean Riemannian geometry**, leading to distorted distances and inefficient exploration of the ambient.
>
> Our framework (PepCompass) is designed to:
> - **correct this mismatch**, and
> - enable both **global navigation** ***(seed discovery)*** and **local optimization** ***(candidate refinement)*** under the correct geometry.
>
> We will make this explicit in the introduction.
>
> ## Motivation for the proposed formulation
>
> The central motivation is the observation that:
> - the **decoder defines a Riemannian geometry** on latent space via its Jacobian (pullback metric).
> - In order for the pullback metric to work out-of-the-box, the Jacobian should be **full-rank** - namely the rank should be equal to the dimensionality of a latent space.
> - However, in practice, the **full-rank manifold assumption often fails**, and we observe this for peptide data.
>
> This leads to a major issue, because the pullback metric becomes **degenerate (rank-deficient)** and hence non-Riemannian. Our formulation (union of κ-stable manifolds) addresses this by:
>
> - Restricting to **locally stable subspaces ($\kappa$-manifolds)** where the Jacobian is full-rank and geometry is well-defined,
> - Putting additional κ-stable condition, that bounds the amount to which metric is distorting the latent space, in order to guarantee a numerical stability,
> - Performing **numerically stable, geometry aware**, Riemannian computations within these subspaces (for SORBES and MUTANG).
>
> We will extend the union of κ-stable manifolds with these intuitions.
>
> ## Intuition behind key components
> We agree that the current presentation is too formal. We will add intuitive explanations for the main components:
>
> #### Union of κ-stable manifolds
> - Instead of assuming one smooth surface, we model peptide space as *many local patches, each with its own valid geometry.*
> - Each patch removes directions where the decoder is insensitive (degenerate directions).
>
> #### SORBES (Riemannian sampling)
> - Standard Gaussian noise uses naive Euclidean distance and ignores decoder-induced geometry.
> - SORBES can be understood as *“Gaussian noise that respects the curvature of the decoder-induced space”.*
>
> #### MUTANG (tangent mutations)
> - Tangent directions indicate **which amino-acid changes matter most**.
> We reinterpret them as *a principled way to generate biologically meaningful mutations.*
>
> #### PoGS (global search)
> Instead of linear interpolation, we search for paths that:
> - stay **geometrically smooth**, w.r.t. to decoder-induced geometry in the peptide space and
> - move toward **high-activity regions**.
>
> We will integrate these intuitions directly into Section 2.
>
> ## Explanation of assumptions (e.g., full-rank assumption)
>
> The full-rank assumption is used in prior work to justify treating the latent space as a smooth manifold with a pullback metric.
>
> However, our empirical findings show that the decoder's Jacobian is often **rank-deficient**, which invalidates this assumption.
>
> This is precisely why we introduce the **union-of-manifolds formulation**.
>
> We will explicitly state this assumption, explain why it fails, and clarify how our method addresses it.
>
> ## Summary of revisions
> In the revised version, we will:
> - Clearly state the **problem and task** in the introduction,
> - Add **high-level intuition** before formal definitions,
> - Explicitly motivate key assumptions and design choices.
>
> We hope these revisions address the reviewer’s concerns and kindly ask them to consider increasing their score in light of these improvements.

---

> > ### Author Rebuttal · Reviewer_vcF5 · 2026-04-03
> >
> > Thank you for the authors’ response. I will increase my score.

---

### Official Review · Reviewer_61YX · 2026-03-13

**Soundness:** 2
**Presentation:** 4
**Significance:** 3
**Originality:** 4
**Overall Recommendation:** 5
**Confidence:** 4

**Summary:**

This paper proposes PepCompass, a geometry-aware framework for the discovery and optimization of AMP. The paper starts from the observation that standard latent-space exploration methods typically treat the latent space as flat Euclidean space, which ignores decoder-induced distortions. It further argues that fixed-dimensional manifold assumptions may fail for peptide generative models. To address this, the authors first introduce a Union of κ-Stable Riemannian Manifolds, and then build two search components on top of it: PoGS for global bi-prototype seed discovery and LE-BO for local optimization using SORBES, MUTANG, and local enumeration. The authors believe that, PoGS yields more seeds/wells than straight interpolation, LE-BO is strong against a broad set of optimization baselines, and the paper includes substantial wet-lab validation on 29 peptides.

**Compliance With Llm Reviewing Policy:**

Affirmed.

**Final Justification:**

This paper makes a strong and practically relevant contribution. I found the method well motivated, the empirical evaluation convincing, and the presentation clear overall. My questions were adequately addressed in the rebuttal, and the authors’ clarifications further strengthened my confidence in the technical soundness of the work.

Overall, I continue to view this as a solid paper with clear relevance to the community. The rebuttal resolved my concerns, and I am maintaining my score of 5.

**Key Questions For Authors:**

Refer to weakness

**Limitations:**

yes

**Strengths And Weaknesses:**

Strengths:
1. The paper is not just another peptide-design benchmark submission. The central claim is that peptide latent spaces are better modeled as a decoder-dependent union of local manifolds rather than a single fixed-dimensional manifold. Algorithms are designed under this central claim. The κ-stable construction is a reasonable way to handle rank deficiency and numerical stability in the pullback metric.
2. PoGS handles global seed discovery by optimizing a path energy that trades off geometric similarity and property potential. SORBES provides local geometry-aware stochastic exploration, while MUTANG translates tangent-space structure into interpretable discrete mutations. LE-BO is then adopted to local enumeration plus BO directly in peptide space instead of directly optimizing an acquisition function in latent space.
3. Wet-lab validation strengthens the paper. The paper reports 29 experimentally tested peptides (4 PoGS seeds + 25 LE-BO analogs). The authors tested against 19 strains including 8 MDR isolates, with a 100% success rate at 32 µg/ml and 82% at 4 µg/ml. This is a major plus for this area.

Weaknesses:
1. PepCompass combines κ-stable manifolds, PoGS, SORBES, MUTANG, local enumeration, GP-based BO, and ROBOT. The main ablations are useful, but I still find it somewhat difficult to cleanly attribute how much of the gain comes from the geometry itself versus better candidate generation, better locality control, or diversity heuristics. Although the experimental results support the whole pipeline, It is not clear what are the minimal essential structures.
2. A key concern is that several central design choices remain heuristic. Although the paper’s main motivation is to move beyond naive Euclidean geometry, the actual implementation still relies on a Euclidean metric in the decoder output space. The authors also acknowledge this as a limitation, and provided reasons, namely that this induces a conservative mutation geometry and penalizes non-local insertions/deletions. Although it is reasonable, it does not fully establish that this is the most natural or biologically appropriate ambient metric. More broadly, components such as the extrinsic chord surrogate in PoGS and the use of stability / threshold hyperparameters suggest that the final system is partly geometry-principled and partly engineering-driven. This does not invalidate the experimental results, but it weaken the strength of the paper’s claim to a fully principled geometric framework.
3. AMP generation, latent-space optimization, BO, pullback-metric geometry, and union-of-manifolds ideas all have prior art. The novelty in this paper is mainly in the combination and adaptation of these ideas to peptide design. The paper should not be oversold as introducing an entirely new research direction from scratch.

---

> ### Author Rebuttal · Authors · 2026-03-30
>
> We thank the reviewer for the insightful feedback. Below, we address each of the concerns in detail.
>
> > W1: PepCompass combines κ-stable manifolds, PoGS, SORBES, MUTANG, local enumeration, GP-based BO, and ROBOT. (..) It is not clear what are the minimal essential structures.
>
> We present the extended ablation of different components (including ROBOT) in Appendix N. Additionally, in Appendix O, we provided a detailed analysis of the impact of geometry-informed exploration on candidate generation. Two key observations emerge: (1) methods that use MUTANG produce a substantially larger number of candidates, (2) a similar trend is observed for approaches using the Riemannian walk. We attribute this effect to improved exploration of high-dimensional regions of the latent space (Appendix Fig. 8). A new analysis of diversity for 5 generated candidates proved it to be higher for LE-BO when compared with its Euclidean counterpart (0.59 vs 0.55 in average normalized Levenshtein distance).
>
> Method| FL14 | KY14 | KK16 | mam-3 | hyd-2 |Avg
> |-|-|-|-|-|-|-|
> |LE-BO|**0.59**|**0.57**|**0.58**|0.60|**0.60**|**0.59**
> |Euc.LE-BO|0.56|0.54|0.54|**0.63**|0.50|0.55
>
>
> > W2.1: A key concern is that several central design choices remain heuristic. Although the paper’s main motivation is to move beyond naive Euclidean geometry, the actual implementation still relies on a Euclidean metric (..)
>
> We agree that while the Euclidean metric provides a mathematically sound, and structurally conservative baseline, it may be biologically simplistic.
> As discussed in Appendix D.3, our framework might be easily extended to biologically informed metrics via the use of mutation kernels. While standard substitution matrices like BLOSUM are natural candidates for such kernels, they present a specific challenge as they reflect broad evolutionary patterns across general protein sequences, which does not necessarily capture the specialized physicochemical and biological intricacies driving antimicrobial activity. To address this, we are actively developing optimization-specific mutation kernels informed by AMP sequences. We view the integration of these biological priors as a highly promising direction for future work.
>
> > W2.2 components such as the extrinsic chord surrogate in PoGS and the use of stability / threshold hyperparameters suggest that the final system is partly geometry-principled and partly engineering-driven. This does not invalidate the experimental results, but it weaken the strength of the paper’s claim to a fully principled geometric framework.
>
> We acknowledge that translating continuous Riemannian geometry to discrete, finite-precision computational models inherently requires certain practical trade-offs. However, bridging this gap between theory and practice is precisely what enables PepCompass to scale to real-world biological data.
>
> Contrary to idealized settings with predefined synthetic manifolds [[1](https://arxiv.org/abs/2311.17289), [2](https://arxiv.org/pdf/2202.00959)] or artificially constrained low-dimensional spaces [[3](https://www.nature.com/articles/s41467-022-29443-w)], modern generative models exhibit complex, dimension-varying latent structure. Furthermore, as they operate under standard `f32` precision, the associated numerical issues cannot be ignored.
>
> Thus, the parameters incorporated into PepCompass serve as principled relaxations to pure theory. For instance, without the κ-stability control in SORBES, inverting the truncated metric in degenerate regions amplifies numerical noise, leading to erratic and unstable latent updates. Setting κ to the machine epsilon approaches the pure geometric interpretation, whereas treating it as a hyperparameter provides the flexibility necessary for robust, real-world applications.
>
> > Q-W3: AMP generation, latent-space optimization, BO, pullback-metric geometry, and union-of-manifolds ideas all have prior art. The novelty in this paper is mainly in the combination and adaptation of these ideas to peptide design. (..)
>
> We agree that our framework builds upon a strong foundation of existing concepts, however, we would like to highlight that it also brings substantial methodological novelties: (1) a principled treatment for the geometrically-informed exploration of data with varying intrinsic dimensionality; (2) a novel interpretation of the continuous tangent space as discrete mutation sets; and (3) a potential-augmented geodesic formulation for targeted property optimization. We believe these contributions go beyond a straightforward application of prior art and introduce a novel paradigm for exploring complex biological spaces.
>
> ---
> We will incorporate these clarifications and additional analyses in the final version of the manuscript to further strengthen the presentation and positioning of our contributions. We hope these revisions address the reviewer’s concerns and look forward to the discussion phase.

---

> > ### Author Rebuttal · Reviewer_61YX · 2026-04-04
> >
> > Thank you for the response, I will keep my score since it is already accepted score.

---

### Decision · Program_Chairs · 2026-04-30

**Decision:**

Accept (regular)

**Comment:**

This paper proposes PepCompass, a geometry aware framework for peptide optimization. PepCompas is a union of riemannian manifolds which the authors argue is a better representation for peptides. PepCompass is experimentally tested in vitro on the anti microbial peptide design task.

Reviewers agree that this work represents an interesting idea backed up by wet-lab validation. I recommend this paper for acceptance at this time.